# Rolling Forcing: Autoregressive Long Video Diffusion in Real Time

**Kunhao Liu**[1][*] **Wenbo Hu**[2][†] **Jiale Xu**[2] **Ying Shan**[2] **Shijian Lu**[1][†]
[1]Nanyang Technological University   [2]ARC Lab, Tencent PCG

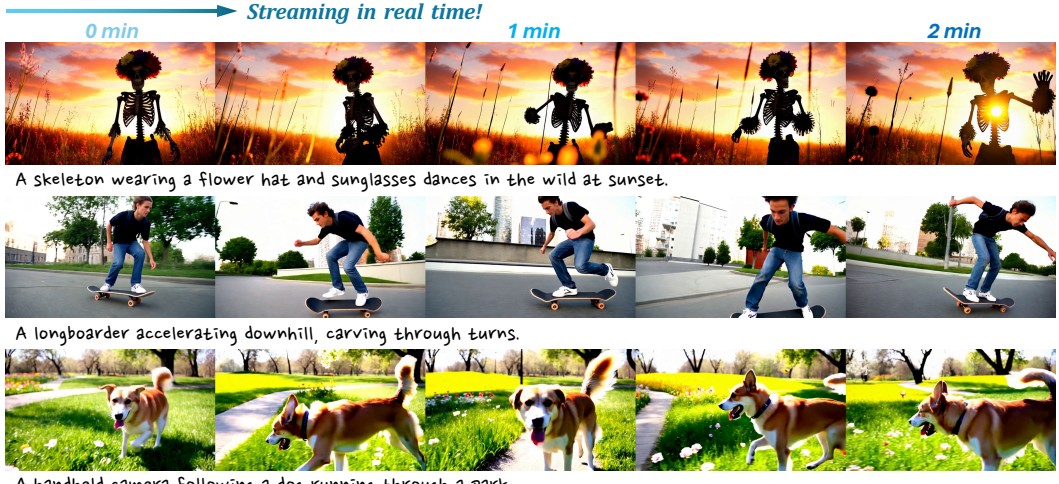

Figure 1: Rolling Forcing performs real-time streaming text-to-video generation at 16 fps on a single GPU and is capable of producing multi-minute-long videos with minimal error accumulation. More results, code, and demo can be found at the project page.

## Abstract

Streaming video generation, as one fundamental component in interactive world models and neural game engines, aims to generate high-quality, low-latency, and temporally coherent long video streams. However, most existing work suffers from severe error accumulation that often significantly degrades the generated stream videos over long horizons. We design Rolling Forcing, a novel video generation technique that enables streaming long videos with minimal error accumulation. Rolling Forcing comes with three novel designs. First, instead of iteratively sampling individual frames, which accelerates error propagation, we design a joint denoising scheme that simultaneously denoises multiple frames with progressively increasing noise levels. This design relaxes the strict causality across adjacent frames, effectively suppressing error growth. Second, we introduce the attention sink mechanism into the long-horizon stream video generation task, which allows the model to keep key–value states of initial frames as a global context anchor and thereby enhances long-term global consistency. Third, we design an efficient training algorithm that enables few-step distillation over largely extended denoising windows. This algorithm operates on non-overlapping windows and mitigates exposure bias conditioned on self-generated histories. Extensive experiments show that Rolling Forcing enables real-time streaming generation of multi-minute videos on a single GPU, with substantially reduced error accumulation.

---

[*]Work done during internship at ARC Lab, Tencent PCG.
[†]Corresponding authors.

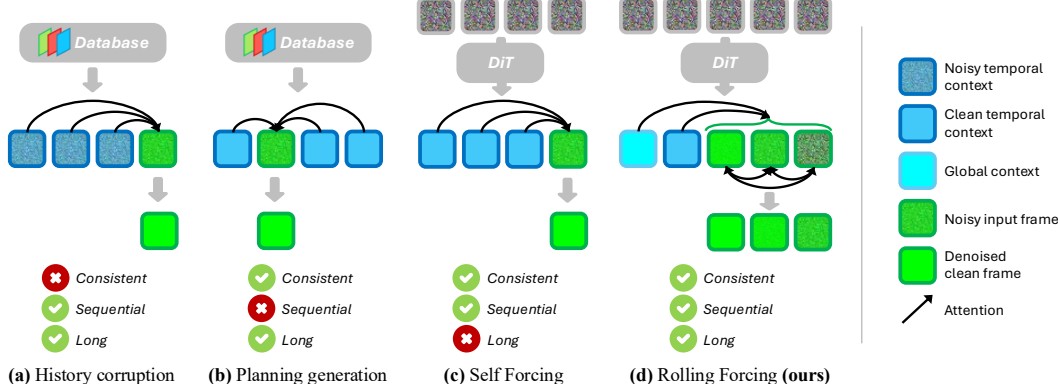

Figure 2: Different paradigms in autoregressive video generation. History corruption (Chen et al., 2024; Guo et al., 2025) in (a) compromises temporal consistency, while planning generation (Zhang & Agrawala, 2025; Xiang et al., 2025) in (b) is incompatible with sequential streaming video generation. Self Forcing (Huang et al., 2025) in (c) can achieve consistent sequential streaming but suffers from severe error accumulation while generating long videos. The proposed Rolling Forcing in (d) supports streaming long video generation with superior temporal consistency and minimal error accumulation.

# 1 INTRODUCTION

Modern video diffusion models (OpenAI, 2024; Polyak et al., 2024) have demonstrated impressive capabilities in generating short video clips with rich detail and coherent motion. However, interactive applications such as world models (Bruce et al., 2024), neural game engines (Valevski et al., 2024), and immersive XR environments require the ability to *stream* each frame with *minimal latency* while maintaining visual quality and temporal coherence over *long horizons*. Unlike offline video generation, where the entire sequence is synthesized together at one go, the streaming video generation operates in an online fashion: frames are generated sequentially and immediately consumed by downstream tasks or displayed to users. Such online nature imposes unique challenges, as the model must maintain long-horizon consistency while accommodating real-time constraints in an autoregressive manner.

Real-time streaming video generation methods, such as CausVid (Yin et al., 2025) and Self Forcing (Huang et al., 2025) (illustrated in Fig. 2(c)), distill a pretrained bidirectional video diffusion model into a fast, causal autoregressive generator. While they enable consistent sequential generation, their strictly causal frame prediction causes each frame to inherit errors from its predecessors, allowing small imperfections to compound over long horizons and eventually leading to noticeable drift and quality degradation. Two representative approaches have been explored for improving video generation over long horizons, as illustrated in Fig. 2(a,b). The first approach explores *history corruption*, which injects noise into past frames to reduce over-reliance on histories (Chen et al., 2024; Guo et al., 2025). History corruption mitigates drift by narrowing the gap between self-generated and ground-truth context, but it deprives the model of clean references and compromises temporal consistency. The second approach explores *planning generation* by first synthesizing distant key frames and then interpolating intermediates (Zhang & Agrawala, 2025; Xiang et al., 2025). Anchoring distant frames to the initial context mitigates drift, but the introduced out-of-order schedule violates strict sequential emission, which is unsuitable for real-time streaming.

We design *Rolling Forcing*, an autoregressive long video generation technique that mitigates error accumulation while maintaining real-time performance as illustrated in Fig. 2. Rolling Forcing comes with three new designs. First, instead of iteratively denoising a single frame at a time as in most existing work, Rolling Forcing introduces rolling-window denoising to process multiple consecutive frames simultaneously. Within each window, frames are connected by bidirectional attention and assigned progressively increasing noise levels. Such mutual refinement corrects local errors before any frame is finalized, thereby suppressing long-horizon drift. In addition, this design

allows us to emit a clean frame after each single forward pass, achieving real-time throughput on a single GPU despite a much larger attention window. Second, we adapt the attention sink mechanism (Xiao et al., 2023) to the streaming video generation task, thereby strengthening long-term global consistency. Specifically, we persist the key–value states of the initial frames as a global context anchor and dynamically adjust their Rotary Position Embeddings (RoPE) (Su et al., 2024), which freezes the relative positions of initial frames to the current denoising frames and prevents excessive offsets. Note that the KV caching is applied to the recent clean frames as well to reduce latency and maintain temporal consistency. Third, we design an efficient training algorithm that enables few-step distillation over the extended denoising windows. This algorithm operates on non-overlapping windows that collectively cover all video frames, mitigating exposure bias by conditioning on self-generated histories during training. Extensive experiments show that Rolling Forcing achieves real-time streaming generation of multi-minute videos on a single GPU, with substantially reduced error accumulation as illustrated in Fig. 1.

The contributions of this work can be summarized in three key aspects. *First*, we introduce a rolling-window joint denoising technique that processes multiple frames in a single forward pass, enabling mutual refinement while preserving real-time latency. *Second*, we introduce the attention sink mechanism into the streaming video generation task, a pioneering effort that enables caching the initial frames as consistent global context for long-term coherence in video generation. *Third*, we design an efficient training algorithm that operates on non-overlapping windows and conditions on self-generated histories, enabling few-step distillation over extended denoising windows and concurrently mitigating exposure bias.

## 2 RELATED WORK

**Bidirectional Video Generation Models.** Video generation has advanced rapidly in recent years, with modern approaches mostly adopting the paradigms of denoising diffusion. Video diffusion has been explored in both pixel space (Ho et al., 2022; Singer et al., 2022) and latent space (Blattmann et al., 2023b;a), with architectures evolving from early Space–Time U-Nets (Blattmann et al., 2023a; Hong et al., 2022) to more recent DiT-based designs (Peebles & Xie, 2023; Gupta et al., 2024). Significant industrial investment has driven the development of large video diffusion models, leading to several multi-billion parameter models, including open-source models such as Wan (Wan et al., 2025) and Hunyuan (Kong et al., 2024), and closed-source models such as Sora (OpenAI, 2024), Movie-Gen (Polyak et al., 2024), and Seaweed (Seawead et al., 2025). Notably, these models operate as bidirectional video diffusion models, as they have access to both past and future frames during denoising. While this bidirectional context enables high-quality synthesis for offline generation, it is incompatible with the causality that is necessitated in real-time streaming video generation.

**Autoregressive Video Generation Models.** To enable long video generation, several studies have extended the generation paradigm from bidirectional to autoregressive, which naturally supports gradual rollout over extended time horizons. Autoregressive models are typically trained with next-token prediction objectives and generate spatiotemporal tokens sequentially at inference time (Bruce et al., 2024; Kondratyuk et al., 2023; Wang et al., 2024; Weissenborn et al., 2019; Yan et al., 2021). More recently, a separate line of research combines autoregressive modeling with denoising diffusion (Chen et al., 2024; Gu et al., 2025; Guo et al., 2025; Jin et al., 2024; Li et al., 2024; Liu et al., 2024; Weng et al., 2024; Yin et al., 2025; Zhang et al., 2025; Zhang & Agrawala, 2025; Huang et al., 2025; Henschel et al., 2025), where frames are generated one-by-one in an outer loop and each frame is gradually denoised in an inner loop. Within this family, Rolling Diffusion (Ruhe et al., 2024) and its variants (Kim et al., 2024; Teng et al., 2025; Sun et al., 2025; Xie et al., 2025; Chen et al., 2025; Teng et al., 2025) merge the outer and inner loops: the diffusion model jointly denoises multiple frames at progressively increasing noise levels. However, these methods mostly suffer from exposure bias and error accumulation when generating long videos. Another line of research addresses error accumulation with planning generation (Long et al., 2024; Zhao et al., 2024; Hu et al., 2024; Xie et al., 2024; Zhang & Agrawala, 2025; Bansal et al., 2024; Yang et al., 2024; Xiang et al., 2025), which predicts distant future frames first and then interpolates the intermediate frames. While effective for reducing drift, it breaks the strict sequential order required for real-time streaming. In contrast, our work enables much longer streaming video generation with minimal error accumulation while addressing exposure bias.

**Concurrent and Closed-Source Work.** Two concurrent works are also devoted to the streaming generation of long videos. Specifically, StreamDiT (Kodaira et al., 2025) adopts the FIFO-style denoising (Kim et al., 2024) for streaming video generation. It modifies the pretrained model architecture by introducing micro-steps and window attention, necessitating extensive additional pretraining with large-scale data and computation. In contrast, our method keeps the pretrained model architecture unchanged, can be trained efficiently in only 3,000 steps, and does not require any video data. APT2 (Lin et al., 2025b) instead explores adversarial distillation (Lin et al., 2025a) for streaming video generation. It denoises videos block-by-block and involves multiple costly post-training stages, including diffusion adaptation, consistency distillation, adversarial training, and long-video training. APT2 is trained on one-minute-long videos, whereas ours is trained only on 5-second clips, yet can extend to multi-minute sequences during inference. Note that both StreamDiT and APT2 are closed-source and trained based on internal video diffusion models (Movie-Gen (Polyak et al., 2024) and Seaweed (Seawead et al., 2025)), while our model is trained on public datasets and relies on an open-source model (*i.e.*, Wan2.1 (Wan et al., 2025)) as its foundation.

## 3 METHODS

### 3.1 PRELIMINARIES: EXPOSURE BIAS IN AUTOREGRESSIVE VIDEO DIFFUSION MODELS

An autoregressive video diffusion model is a hybrid generative framework that integrates autoregressive chain-rule decomposition with denoising diffusion for video generation. Formally, given a sequence of $N$ video frames $x^{1:N} = (x^1, x^2, \ldots, x^N)$, their joint distribution can be factorized using the chain rule: $p(x^{1:N}) = \prod_{i=1}^{N} p(x^i \mid x^{<i})$. Each conditional distribution $p(x^i \mid x^{<i})$ is modeled through a diffusion process, where each frame is generated by progressively denoising Gaussian noise while conditioning on the previously generated frames. In practice, one may also generate a chunk of consecutive frames instead of a single frame at each step (Yin et al., 2025; Teng et al., 2025; Huang et al., 2025). For clarity, we refer to each chunk simply as a frame in the following text.

Autoregressive video diffusion models are trained either (1) from scratch with frame-wise denoising loss or (2) by distilling a pretrained bidirectional model. The first approach is trained under the paradigm of Teacher Forcing (TF) or Diffusion Forcing (DF) (Chen et al., 2024). In TF, the conditional distribution for the $i$th frame at noise level $t_j$ is $p(x^i_{t_j} \mid x^{<i}_0)$, where all conditional history frames are the ground-truth clean frames from the training data. While in DF, the conditional distribution is $p(x^i_{t_j} \mid x^{<i}_{t_{\geq 0}})$, where the history frames are the ground-truth frames corrupted with independent noise levels. Since training relies on ground-truth histories while inference relies on the model's own predictions, a train–test gap known as exposure bias arises (Schmidt, 2019). Mitigating the exposure bias is difficult because the denoising loss requires pairs of model predictions and the corresponding ground truth conditioned on them, which are unavailable.

The second approach of distillation, however, provides a way to bypass the denoising loss and mitigate exposure bias. CausVid (Yin et al., 2025) distills a pretrained bidirectional model into a few-step causal model. It adopts a Distribution Matching Distillation (DMD) loss (Yin et al., 2024b) that minimizes the reverse KL divergence across randomly sampled timesteps $t$ between the smoothed data distribution $p_{\text{data}}(x_t)$ and the student generator's output distribution $p_{\text{gen}}(x_t)$. The gradient of the reverse KL can be approximated as the difference between two score functions:

$$\nabla_\theta \mathcal{L}_{\text{DMD}} \triangleq \mathbb{E}_t \left( \nabla_\theta \text{KL} \left( p_{\text{gen},t} \| p_{\text{data},t} \right) \right)$$
$$\approx -\mathbb{E}_t \left( \int \left( s_{\text{data}} \left( \Psi \left( G_\theta(\epsilon), t \right), t \right) - s_{\text{gen}} \left( \Psi \left( G_\theta(\epsilon), t \right), t \right) \right) \frac{dG_\theta(\epsilon)}{d\theta} \, d\epsilon \right), \quad (1)$$

where $\Psi$ represents the forward diffusion process, $\epsilon$ is random Gaussian noise, $G_\theta$ is the generator parameterized by $\theta$, and $s_{\text{data}}$ and $s_{\text{gen}}$ represent the score functions trained on the data and generator's output distribution, respectively. Since training with DMD loss does not require ground-truth image or video data (Yin et al., 2024a), Self Forcing (Huang et al., 2025) mitigates the exposure bias by conditioning each frame on previously self-generated histories during training. However, although exposure bias is alleviated, severe error accumulation still occurs once generation extends beyond the trained temporal window.

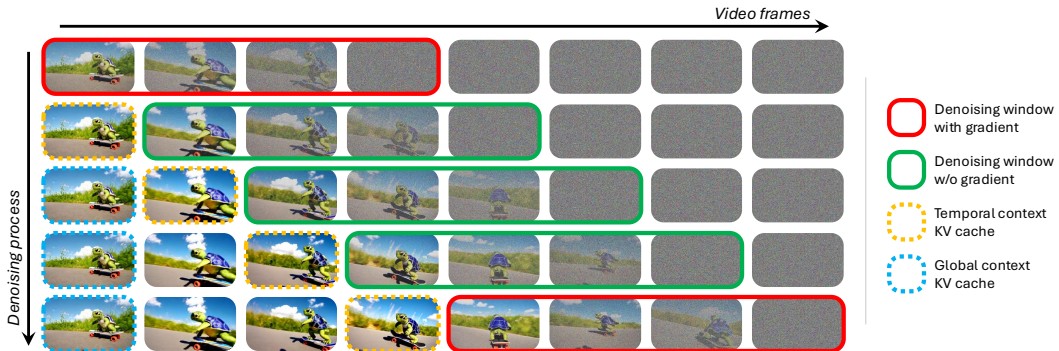

Figure 3: Illustration of the Rolling Forcing denoising process with $T = 4$. Rolling Forcing jointly denoises a short window of consecutive frames that are assigned progressively higher noise levels and connected by bidirectional attention. The KV cache of recent frames is preserved as temporal context to maintain short-term consistency, while the KV cache of the initial frames is preserved as global context to ensure long-term consistency. During training, only a subset of denoising windows requires gradient computation, as highlighted by the red windows. These windows are mutually exclusive yet collectively cover all video frames.

## 3.2 AUTOREGRESSIVE VIDEO GENERATION VIA ROLLING DIFFUSION WINDOW

In Self Forcing (SF), videos are generated frame-by-frame in a strict causal manner. Consider a noise schedule $\{t_0 = 0, t_1, \dots, t_T = 1000\}$ with total noise levels $T + 1$. At each denoising step $t_j$ and frame index $i$, the model denoises an intermediate noisy frame $x_{t_j}^i$ conditioned on previous clean frames $x_0^{<i}$ and then injects Gaussian noise with a lower noise level into the predicted denoised clean frame via the forward diffusion process $\Psi$. This produces a noisy frame $x_{t_{j-1}}^i$ which will be used as the input to the next denoising step. Formally, in SF, the denoising process is achieved by: $x_{t_{j-1}}^i = \Psi\big(G_\theta(x_{t_j}^i, t_j, x_0^{<i}), t_{j-1}\big)$, and $x_{t_T}^i \sim \mathcal{N}(0, I)$. However, this formulation has no bidirectional attention between the current denoising frame $x^i$ and its history $x^{<i}$, where the strict causality forces every frame to inherit and compound the errors from its predecessors over time.

The proposed Rolling Forcing relaxes this constraint by extending the single-frame denoising window into a rolling window spanning multiple frames, as illustrated in Fig. 3. Each denoising window contains consecutive frames with progressively higher noise levels in temporal order, akin to Rolling Diffusion (Ruhe et al., 2024). The length of the denoising window $L_{\text{win}}$ is set to the number of denoising time steps, *i.e.*, $L_{\text{win}} = T$. To ensure continuity, the next noise level of the $i$th frame is aligned with the current noise level of the $(i - 1)$th frame, allowing the window to roll forward infinitely. At each roll, a clean frame is generated, and pure Gaussian noise is appended as the next frame to be synthesized. Formally, for the denoising window starting at the $i$th frame, the denoising distribution of Rolling Forcing can be defined by:

$$p_\theta\Big(x_{t_{0:T-1}}^{i:i+T-1} \mid x_{t_{1:T}}^{i:i+T-1}, x_0^{<i}\Big) = \Psi\Big(G_\theta(x_{t_{1:T}}^{i:i+T-1}, t_{1:T}, x_0^{<i}), t_{0:T-1}\Big), \quad (2)$$

where $x_{t_{1:T}}^{i:i+T-1}$ denotes the noisy frames in the denoising window, and $x_{t_{0:T-1}}^{i:i+T-1}$ denotes the window output with each frame denoised to a lower noise level. The generator $G_\theta$ predicts clean frames conditioned on the input noisy frames, their noise levels $t_{1:T}$, and the clean history frames $x_0^{<i}$. $\Psi$ injects Gaussian noise $\epsilon_{t_{0:T-1}}$ at noise levels $t_{0:T-1}$ into the predicted clean frames, producing frames with reduced noise levels.

Since the length of the denoising window equals the number of denoising steps $T$, which is typically large (*i.e.*, $\sim 50$) in video diffusion models (Wan et al., 2025), the denoising window itself becomes prohibitively large. To manage such large windows, previous work either processes every frame independently on multiple GPUs (Kim et al., 2024), or reduces $T$ to $\sim 30$ using few-step samplers (Xie et al., 2025). In contrast, we adopt diffusion distillation (Yin et al., 2024b;a), which reduces the number of denoising steps $T$ to just 5 while preserving generation quality, thereby making the denoising windows compact enough to fit on a single GPU while maintaining real-time latency.

### 3.3 TEMPORAL AND GLOBAL HISTORY CONTEXT

As the clean history frames $x_0^{<i}$ accumulate during generation, handling them directly becomes computationally expensive. To address this, following Huang et al. (2025), we cache the key and value states of the history frames, thereby avoiding redundant recomputation when generating new frames, as illustrated in Fig. 3. Note that although the attention within the denoising window is bidirectional, the attention between the frames in the denoising window and the KV cache of history frames remains causal. While KV caching reduces computation, the computational complexity still grows quadratically with the cache size as frames accumulate, and the cache may become large enough to cause out-of-memory errors. Given a denoising window starting at the $i$th frame $x_{t_{1:T}}^{i:i+T-1}$, we address this issue by retaining only the KV cache of the most recent $L_{\text{tem}}$ history frames $x_0^{i-L_{\text{tem}}:i-1}$ as temporal context to preserve short-term temporal consistency. However, relying solely on short-term history causes a gradual drift of long-range properties of the generated video (like exposure, color tone, white balance, etc.) as generation proceeds.

To maintain long-term global consistency, we cache the KV states of the initial $L_{\text{glo}}$ generated frames $x_0^{1:L_{\text{glo}}}$ as global context, analogous to at-

---

**Algorithm 1** Rolling Forcing Training

**Require:** Denoise timesteps $\{t_0, t_1, \ldots, t_T\}$
**Require:** Number of video frames $N$
**Require:** AR diffusion model $G_\theta$ (returns KV embeddings via $G_\theta^{\text{KV}}$)

1: **loop**
2:     Initialize model output $\mathbf{X}_\theta \leftarrow [\,]$
3:     Initialize KV cache $\mathbf{KV} \leftarrow [\,]$
4:     Initialize $x_{t_{1:T-1}}^{1:T-1}$ with $G_\theta$
5:     Sample $j \sim \text{Uniform}\{0, 1, \ldots, T-1\}$
6:     **for** $i = 1, \ldots, N$ **do**
7:         Sample $x_{t_T}^{i+T-1} \sim \mathcal{N}(0, I)$
8:         Set $x_{t_{1:T}}^{i:i+T-1} \leftarrow x_{t_{1:T-1}}^{i:i+T-2} \,\|\, x_{t_T}^{i+T-1}$
9:         Select and apply RoPE to $\mathbf{KV}$ (Sec. 3.3)
10:        **if** $i \equiv j \pmod{T}$ **then**
11:           Enable gradient computation
12:           $\hat{x}_0^{i:i+T-1} \leftarrow G_\theta(x_{t_{1:T}}^{i:i+T-1}, t_{1:T}, \mathbf{KV})$
13:           $\mathbf{X}_\theta\,.\,\texttt{append}(\hat{x}_0^{i:i+T-1})$
14:           Disable gradient computation
15:        **else**
16:           $\hat{x}_0^{i:i+T-1} \leftarrow G_\theta(x_{t_{1:T}}^{i:i+T-1}, t_{1:T}, \mathbf{KV})$
17:        **end if**
18:        $\mathbf{KV}\,.\,\texttt{append}(G_\theta^{\text{KV}}(\hat{x}_0^i, t_0, \mathbf{KV}))$
19:        $x_{t_{1:T-1}}^{i+1:i+T-1} \leftarrow \Psi(\hat{x}_0^{i+1:i+T-1}, t_{1:T-1})$
20:     **end for**
21:     Update $\theta$ via DMD loss (Eq. (1))
22: **end loop**

---

tention sink tokens in streaming language models (Xiao et al., 2023). The cache sizes $L_{\text{tem}}$ and $L_{\text{glo}}$ are chosen such that the total attention window size matches that of the bidirectional teacher model, *i.e.*, $L_{\text{tem}} + L_{\text{glo}} + L_{\text{win}} = L_{\text{bidirectional}}$. However, directly caching the initial frames leads to spilling problems. Modern video diffusion DiTs (Peebles & Xie, 2023) typically use RoPE (Su et al., 2024) for relative positional encoding. As the indices of the denoising frames $i : i + T - 1$ increase, their relative distance to the initial cached frames grows, eventually exceeding the trained range of RoPE and producing unnatural artifacts. To resolve this, we cache the key states of the global context frames $x_0^{1:L_{\text{glo}}}$ before applying the RoPE transformation. During generation, we dynamically apply RoPE to these cached key states at the effective indices $i - L_{\text{tem}} - L_{\text{glo}} : i - L_{\text{tem}} - 1$, treating them as being positioned immediately before the temporal context frames $x_0^{i-L_{\text{tem}}:i-1}$. This adjustment preserves a fixed relative position w.r.t. the denoising frames, preventing excessive offsets.

### 3.4 ROLLING FORCING POST-TRAINING

Rolling Forcing distills a pretrained bidirectional video diffusion model (Wan et al., 2025) to a few-step causal autoregressive generator using the DMD loss (Eq. (1)). As DMD matches the holistic distribution of the entire video sequence to the data distribution $D(p_{\text{data}}(x^{1:N}) \| p_\theta(x^{1:N}))$, the calculation of the DMD loss requires a predicted clean video $\hat{x}_0^{1:N}$ during training. In SF, the predicted clean video is generated by:

$$\hat{x}_0^{1:N} = \left\{ \hat{x}_0^i = G_\theta(x_{t_j}^i, t_j, x_0^{<i}) \mid i = 1, 2, \ldots, N \right\}, \tag{3}$$

where $j \sim \text{Uniform}\{0, 1, \ldots, T-1\}$ indicates each frame's noise level $t_j$ before denoising. For Rolling Forcing, as the denoising window consists of multiple frames at different noise levels, we select the $j$th frame in each window and combine the selected frames as the predicted clean video:

$$\hat{x}_0^{1:N} = \left\{ \hat{x}_0^i = (\hat{x}_0^{i:i+T-1})^j = G_\theta(x_{t_{1:T}}^{i:i+T-1}, t_{1:T}, x_0^{<i})^j \mid i = 1, 2, \ldots, N \right\}, \tag{4}$$

where $j \sim \text{Uniform}\{0, 1, \ldots, T-1\}$ represents both the frame's index within the denoising window and the frame's noise level $t_j$. However, Eq. (4) incurs $T$ times higher computational complexity

Table 1: Comparisons with relevant baselines. We compare Rolling Forcing with representative open-source autoregressive video generation models.

| Model | #Params | Throughput (FPS) ↑ | Latency (s) ↓ | Temporal Flickering | Subject Consistency | Background Consistency | Motion Smoothness | Aesthetic Quality | Imaging Quality | $\Delta$ Quality Drift ↓ |
|---|---|---|---|---|---|---|---|---|---|---|
| *Non-Streaming* | | | | | | | | | | |
| FramePack (Zhang & Agrawala, 2025) | 13B | 0.92 | 65 | **99.26** | 91.65 | 93.55 | 99.03 | 59.93 | 65.20 | 3.45 |
| *Diffusion Forcing Causal* | | | | | | | | | | |
| SkyReels-V2 (Chen et al., 2025) | 1.3B | 0.49[†] | 112[†] | 97.43 | 89.23 | 93.45 | 98.76 | 61.55 | 62.90 | 5.59 |
| MAGI-1 (Teng et al., 2025) | 4.5B | 0.19[†] | 282[†] | 98.21 | 90.86 | 93.25 | **99.20** | 59.91 | 59.87 | 2.15 |
| *Distilled Causal* | | | | | | | | | | |
| CausVid (Yin et al., 2025) | 1.3B | 15.38 | 0.78 | 96.84 | 87.99 | 89.99 | 98.09 | 60.95 | 66.38 | 2.18 |
| Self Forcing (Huang et al., 2025) | 1.3B | 15.38 | 0.78 | 97.49 | 86.48 | 90.29 | 98.47 | 60.54 | 68.68 | 1.66 |
| Rolling Forcing (Ours) | 1.3B | **15.79** | **0.76** | 97.61 | **92.80** | **93.71** | 98.70 | **62.39** | **70.75** | **0.01** |

[†] Numbers adopted from Huang et al. (2025).

than Eq. (3), because the query size is $T$ times larger. Given that DMD loss is already computationally expensive, this additional cost can easily lead to out-of-memory error even on GPUs with 80G of memory.

To address this issue, instead of backpropagating through every window (which requires gradients for each forward pass), we sample a subset of non-overlapping windows to construct the predicted clean video, as illustrated in Fig. 3. Gradient computation is performed only on these selected windows, significantly reducing memory usage while retaining effective supervision. Formally, the predicted clean video is given by:

$$\hat{x}_0^{1:N} = \left\{ \hat{x}_0^{i:i+T-1} = G_\theta(x_{t_{1:T}}^{i:i+T-1}, t_{1:T}, x_0^{<i}) \mid i \equiv j \pmod{T},\ 1 \le i \le N \right\}, \tag{5}$$

where $j \sim \text{Uniform}\{0, 1, \ldots, T-1\}$. In each iteration, we reduce the number of forward passes requiring gradient computation from $N$ in Eq. (4) to $\lceil N/T \rceil$[1]. The Rolling Forcing training is illustrated in Alg. 1. Similar to SF, the input noisy frames $x_{t_{1:T}}^{i:i+T-1}$ during training are generated by the model rather than taken from ground truth, thus mitigating the exposure bias. However, unlike Eq. (3) or Eq. (4), where every frame in the predicted clean video $\hat{x}_0^{1:N}$ is denoised from the same noise level $t_j$, the frames in Eq. (5) are denoised from varying noise levels $t_{1:T}$. Consequently, frames denoised from different noise levels have different quality and clarity, leading to unnatural video $\hat{x}_0^{1:N}$ and camera movement in DMD training. To address this issue, we adopt a mixed training strategy that alternates between SF training (Eq. (3)) and Rolling Forcing training (Eq. (5)) with equal probability. The SF objective serves as a regularizer, encouraging the model to produce videos with natural camera movement. The inference adopts the Rolling Forcing paradigm alone as elaborated in Alg. 2.

## 4 EXPERIMENTS

### 4.1 IMPLEMENTATION DETAILS

**Model.** We implement Rolling Forcing with Wan2.1-T2V-1.3B (Wan et al., 2025) as our base model, which generates 5s videos at 16 FPS with a resolution of $832 \times 480$. Following CausVid (Yin et al., 2025) and Self Forcing (Huang et al., 2025), we first initialize the base model with causal attention masking on 16k ODE solution pairs sampled from the base model. For both ODE initialization and Rolling Forcing training, we sample text prompts from a filtered and LLM-extended version of VidProM (Wang & Yang, 2024). We set $T = 5$ and perform chunk-wise denoising with each chunk containing 3 latent frames. The model is trained for 3,000 steps with a batch size of 8 and a trained temporal window of 27 latent frames. We use the AdamW optimizer for both the generator $G_\theta$ (learning rate $1.5 \times 10^{-6}$) and the fake score $s_{\text{gen}}$ (learning rate $4.0 \times 10^{-7}$). The generator is updated every 5 steps of fake score updates.

**Evaluation.** We adopt the VBench (Huang et al., 2024) quality matrices to evaluate the generation quality over 200 randomly sampled MovieGen (Polyak et al., 2024) prompts, where the matrices

---

[1]We omit the denoising windows at the start of the video for clarity in Eqs. (2) and (5), where the window has fewer than $T$ frames. Gradient computation is still required if the window index $i$ satisfies $i \equiv j \pmod{T}$.

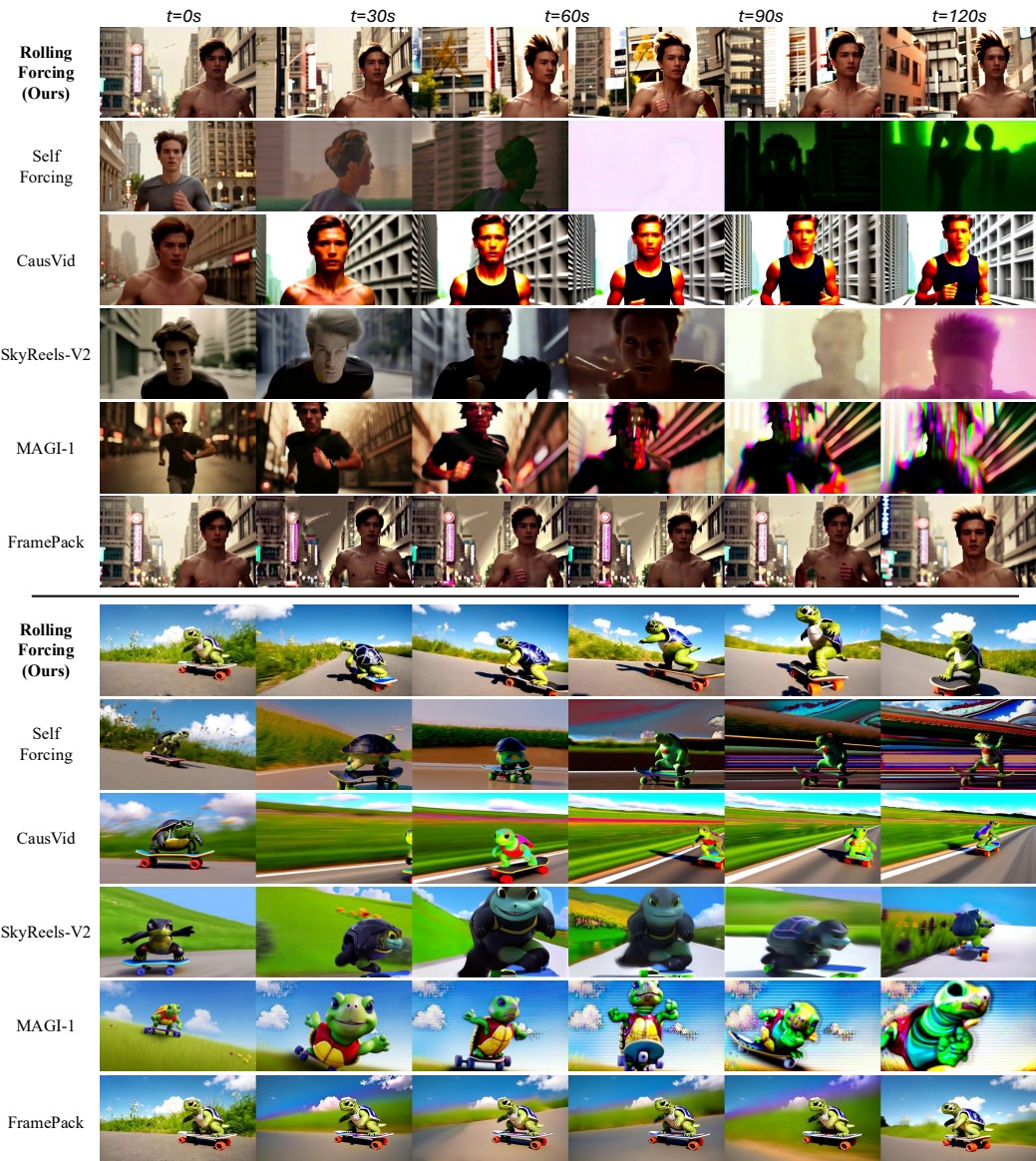

Figure 4: Qualitative comparisons. We compare Rolling Forcing with representative open-source autoregressive video generation models on long video generation.

measure multiple dimensions, including temporal flickering, subject consistency, background consistency, motion smoothness, aesthetic quality, and imaging quality. For fairness, all videos for quantitative evaluation are generated with the same length (30s), frame rate (16 fps), and resolution ($832 \times 480$). To assess quality drift in long video generation, following Zhang & Agrawala (2025); Yin et al. (2025), we compute the absolute difference in imaging quality, $\Delta_{\text{Drift}}^{\text{Quality}}$, between the first and the last 5 seconds of each video. The magnitude of $\Delta_{\text{Drift}}^{\text{Quality}}$ directly reflects the severity of error accumulation. Following Huang et al. (2025), we evaluate real-time performance in terms of both throughput and latency. Unlike prior work that reports the first-frame latency, we measure latency after the generation process reaches a stable speed.

## 4.2 COMPARISONS

We compare Rolling Forcing against several relevant open-source video generation models of comparable scale. Specifically, SkyReels-V2 (Chen et al., 2025) is trained under the Diffusion Forcing paradigm (Chen et al., 2024), which corrupts historical frames during inference to alleviate error

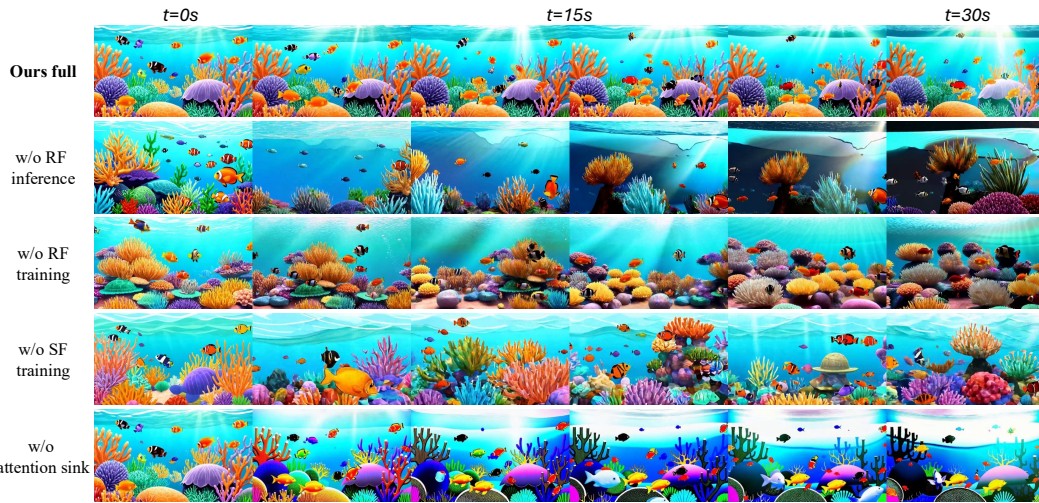

Figure 5: Ablation studies on rolling diffusion window, mixed training strategy, and attention sink.

accumulation. MAGI-1 (Teng et al., 2025) adopts a FIFO-style denoising paradigm (Kim et al., 2024) in both training and inference. We also compare against prior distillation-based approaches, including CausVid (Yin et al., 2025) and Self Forcing (Huang et al., 2025). Note that SkyReels-V2, CausVid, Self Forcing, and our Rolling Forcing are all initialized from the same base model, Wan2.1-T2V-1.3B (Wan et al., 2025). Additionally, we compare with the non-streaming image-to-video method FramePack (Zhang & Agrawala, 2025), which generates long videos in a non-sequential order. We use the first frame generated by our method as the image input for FramePack.

As shown in Table 1, Rolling Forcing achieves the highest overall quality scores. In particular, it obtains a substantially lower $\Delta_{\text{Drift}}^{\text{Quality}}$, demonstrating its effectiveness in suppressing error accumulation. Qualitative comparisons in Fig. 4 further highlight that Rolling Forcing preserves high-fidelity and consistent video quality over 2 minutes of autoregressive generation, while the compared models exhibit pronounced degradation, such as color shifts, artifacts, unnatural motion, etc. In addition, Rolling Forcing achieves real-time generation with sub-second latency, marginally faster than Self Forcing and CausVid, thereby establishing its suitability for long-horizon video streaming applications. Notably, Rolling Forcing even outperforms the non-streaming method FramePack in quality, consistency, and drift control, despite FramePack being a 10 times larger image-to-video model. As shown in Fig. 4, FramePack still suffers from severe error accumulation when generating long videos and tends to generate static outputs. Note that FramePack generates videos in reverse order, thus error accumulation is most severe at the beginning of the videos.

## 4.3 ABLATION STUDIES

We conduct ablation studies to assess the contribution of several design options, as summarized in Table 2.

**Rolling diffusion window.** We evaluate two variants: w/o RF inference and w/o RF training. In w/o RF inference, we remove the rolling denoising window and adopt frame-by-frame denoising during inference,

Table 2: Ablation studies. RF refers to Rolling Forcing, and SF refers to Self Forcing.

| Model | Evaluation Scores ↑ | | | | | | $\Delta_{\text{Drift}}^{\text{Quality}}$ ↓ |
|---|---|---|---|---|---|---|---|
| | Temp. | Subj. | Back. | Mot. | Aes. | Img. | |
| w/o RF inference | 95.45 | 86.01 | 89.94 | 97.36 | 57.59 | 65.19 | 5.53 |
| w/o RF training | 95.91 | 87.50 | 90.86 | 98.05 | 60.41 | 69.24 | 0.89 |
| w/o SF training | 90.83 | 83.27 | 88.14 | 95.63 | 55.30 | 62.00 | 1.62 |
| w/o attention sink | 97.53 | 83.22 | 87.99 | 98.56 | 58.99 | 67.30 | 4.63 |
| Ours full | **97.61** | **92.80** | **93.71** | **98.70** | **62.39** | **70.75** | **0.01** |

while keeping the same training procedure and model weights as our full method. In w/o RF training, the model is trained and inferred entirely with the frame-by-frame paradigm. As shown in Fig. 5, both variants suffer from noticeable error accumulation within 30s, demonstrating that the rolling window is crucial for suppressing long-term drift.

**Mixed training strategy.** To assess its effect, we remove the Self Forcing training objective (w/o SF training). As reported in Table 2, this leads to substantial degradation in consistency and overall quality, primarily due to unnatural camera motion.

**Attention sink.** Finally, removing the global context frame (w/o attention sink) results in noticeable drift in the generated videos, as illustrated in Fig. 5.

## 5 CONCLUSION

We presented *Rolling Forcing*, a framework for real-time long-horizon video generation that mitigates error accumulation while sustaining sub-second latency. By introducing a rolling-window joint denoising strategy, Rolling Forcing enables mutual refinement across consecutive frames, effectively reducing long-term drift. The integration of the attention sink mechanism further enhances global consistency by anchoring initial frames as persistent context, while our efficient training algorithm enables few-step distillation over extended denoising windows while mitigating exposure bias. Extensive experiments demonstrate that Rolling Forcing achieves state-of-the-art temporal coherence and visual fidelity over multi-minute streaming sequences, significantly outperforming prior streaming approaches in both quality and efficiency.

## ACKNOWLEDGEMENT

This project is funded by the Ministry of Education Singapore, under the Tier-1 project scheme with project number RT18/22.

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

## A  ADDITIONAL IMPLEMENTATION DETAILS

**KV Cache.**  We configure the KV cache and denoising window sizes as $L_{\text{tem}} = 3$, $L_{\text{glo}} = 3$, and $L_{\text{win}} = 15$ latent frames. When updating the KV cache with $G_\theta^{\text{KV}}$, the attention window attends only to recent frames, excluding the global context. This design reflects that, apart from the initial frames serving as the global context anchor, other cached frames are retained solely for preserving short-term temporal consistency. During inference, we persist the KV states of the global context frames while discarding obsolete temporal frames, thereby maintaining constant memory usage. At the start of the video, when the denoising window still includes the first frame, no temporal or global context is used.

**Training.**  During training, the number of generated frames $N$ is randomly sampled between 21 (the sequence length of the bidirectional teacher

---

**Algorithm 2** Rolling Forcing Inference

**Require:** Denoise timesteps $\{t_0, t_1, \ldots, t_T\}$
**Require:** Number of video frames $N$
**Require:** AR diffusion model $G_\theta$ (returns KV embeddings via $G_\theta^{\text{KV}}$)
1: Initialize model output $\mathbf{X}_\theta \leftarrow []$
2: Initialize KV cache $\mathbf{KV} \leftarrow []$
3: Initialize $x_{t_{1:T-1}}^{1:T-1}$ with $G_\theta$
4: **for** $i = 1, \ldots, N$ **do**
5: $\quad$ Sample $x_{t_T}^{i+T-1} \sim \mathcal{N}(0, I)$
6: $\quad$ Set $x_{t_{1:T}}^{i:i+T-1} \leftarrow x_{t_{1:T-1}}^{i:i+T-2} \parallel x_{t_T}^{i+T-1}$
7: $\quad$ Select and apply RoPE to $\mathbf{KV}$ (Sec. 3.3)
8: $\quad \hat{x}_0^{i:i+T-1} \leftarrow G_\theta(x_{t_{1:T}}^{i:i+T-1}, t_{1:T}, \mathbf{KV})$
9: $\quad \mathbf{X}_\theta.\texttt{append}(\hat{x}_0^i)$
10: $\quad \mathbf{KV}.\texttt{append}(G_\theta^{\text{KV}}(\hat{x}_0^i, t_0, \mathbf{KV}))$
11: $\quad x_{t_{1:T-1}}^{i+1:i+T-1} \leftarrow \Psi(\hat{x}_0^{i+1:i+T-1}, t_{1:T-1})$
12: **end for**

---

model) and 27 latent frames.  The DMD loss is computed on the last 21 frames.  Since in Wan2.1 (Wan et al., 2025) the first VAE-encoded frame is not temporally compressed and thus exhibits different statistics, we decode frames $0{:}N{-}21$ to RGB and re-encode the $(N{-}21)$-th frame into the latent space. This re-encoded frame is then concatenated with latent frames $N{-}21{:}N{-}1$ for loss computation. In this way, the first frame is only spatially compressed, ensuring consistency with the statistical distribution of the bidirectional teacher model.

**Noise schedule and model parameterization.**  Following the Wan2.1 and Self Forcing, we adopt the flow matching framework (Lipman et al., 2022; Liu et al., 2022), with time step shifting $t'(k, t) = (kt/1000)/(1 + (k - 1)(t/1000)) \cdot 1000$ and a shift factor $k = 5$. The forward process is specified as $x_t = \frac{t'}{1000}x + \frac{1-t'}{1000}\epsilon, \epsilon \sim \mathcal{N}(0, I)$ with $t \in [0, 1000]$. The data prediction model is given by:

$$G_\theta(x, t, c) = c_{\text{skip}} \cdot \epsilon - c_{\text{out}} \cdot v_\theta(c_{\text{in}} \cdot x_t, c_{\text{noise}}(t'), c). \tag{6}$$

We keep the preconditioning coefficients the same as the base models' configuration, i.e., $c_{\text{skip}} = c_{\text{in}} = c_{\text{out}} = 1$ and $c_{\text{noise}}(t) = t$. Our few-step diffusion process employs a uniform 5-step schedule $[t_5, t_4, t_3, t_2, t_1] = [1000, 800, 600, 400, 200]$.  We adopt a 5-step schedule rather than 4, as our method achieves comparable and even slightly faster generation speed than 4-step Self Forcing.

**Indices of the 200 sampled MovieGen prompts.**  0, 5, 10, 15, 24, 30, 34, 38, 44, 48, 53, 60, 67, 71, 75, 79, 84, 88, 92, 98, 103, 108, 112, 116, 122, 126, 131, 137, 142, 146, 150, 157, 165, 171, 176, 182, 188, 196, 200, 207, 211, 215, 219, 225, 229, 233, 237, 242, 246, 250, 256, 261, 267, 272, 277, 284, 288, 294, 299, 303, 308, 312, 317, 321, 327, 331, 336, 340, 344, 348, 353, 357, 362, 367, 372, 376, 380, 388, 392, 396, 400, 408, 415, 423, 428, 433, 437, 441, 446, 452, 456, 460, 464, 469, 473, 477, 482, 487, 491, 495, 502, 507, 511, 515, 521, 525, 529, 533, 540, 544, 548, 553, 558, 569, 574, 578, 585, 590, 598, 602, 609, 614, 619, 626, 632, 636, 641, 647, 651, 657, 661, 666, 671, 677, 681, 686, 690, 695, 699, 704, 708, 712, 717, 722, 726, 730, 734, 739, 743, 747, 752, 756, 761, 766, 772, 776, 781, 786, 791, 795, 799, 803, 808, 812, 816, 820, 825, 829, 834, 838, 845, 849, 855, 860, 865, 870, 875, 880, 884, 888, 892, 897, 904, 908, 915, 924, 928, 933, 937, 942, 946, 954, 959, 964, 970, 976, 980, 986, 991, 996.

## B  INTERACTIVE VIDEO STREAMING

In Fig. 6, we demonstrate that Rolling Forcing enables interactive video streaming, allowing users to modify prompts during generation to steer the video content. Implementing this functionality is straightforward: we discard the cross-attention cache of previous text prompts and apply the new prompts in cross-attention. Rolling Forcing is able to perform various changes including subject switch, subject introduction, scene switch, and action switch.

**Subject Switch**

A horse is running in desert. -> A Tiger is running in desert.

A skeleton is dancing. -> A man is dancing.

**Subject Introduction**

A vast mountain valley at morning. -> A sleek futuristic spacecraft descends over the ridge.

A brown puppy sits on riverbank pebbles beside a river. -> A little girl in a light blue dress walks to the dog.

**Scene Switch**

A young male longboarder accelerating downhill. -> The longboarder slides into the forest.

A skier racing down a steep slope. -> Stars and moons swirling around the skier.

**Action Switch**

An older man playing the piano. -> The old man is eating a burger.

A young woman with a bright smile. -> The woman is waving her hands enthusiastically.

Figure 6: Interactive Video Streaming. Rolling Forcing allows the users to change prompts while streaming to steer the video content.

## C    EVALUATIONS ON 2-MINUTE VIDEOS

To further evaluate our method's performance in long video generation, we compare it with real-time baselines, *i.e.* CausVid (Yin et al., 2025) and Self Forcing (Huang et al., 2025), on 2-minute videos. As shown in Table 3, Rolling Forcing demonstrates a more pronounced advantage in this setting, outperforming both baselines across nearly all quality metrics and exhibiting significantly lower quality drift. We also analyze the CLIP score (Radford et al., 2021) for each time segment over the 2-minute generated videos in Fig. 7. The results show that Rolling Forcing achieves the highest scores across segments, with the smallest score drop over time.

Table 3: Evaluations on 2-minute videos. We compare Rolling Forcing with real-time baselines and w/o RF training variant.

| Model | Evaluation Scores ↑ | | | | | | | $\Delta_{\text{Drift}}^{\text{Quality}}$ ↓ |
|---|---|---|---|---|---|---|---|---|
| | Temp. | Subj. | Back. | Mot. | Aes. | Img. | Dyn. | |
| CausVid (Yin et al., 2025) | 96.67 | 84.69 | 89.53 | 98.04 | 62.16 | 63.62 | 52.08 | 3.35 |
| Self Forcing (Huang et al., 2025) | **97.44** | 71.95 | 88.73 | 98.19 | 50.66 | 60.03 | 51.02 | 14.4 |
| Ours w/o RF training | 96.87 | 84.38 | 91.93 | **98.39** | 58.61 | 66.55 | 56.75 | 1.39 |
| Rolling Forcing (Ours) | 96.90 | **91.47** | **95.29** | 98.29 | **65.21** | **68.96** | **57.14** | **0.49** |

To further demonstrate the effectiveness of the rolling denoising window strategy, we conduct an ablation study on 2-minute videos by removing Rolling Forcing (RF) during training, where the model is trained and inferred entirely under the frame-by-frame paradigm. As shown by the quantitative metrics in Table 3, removing the rolling denoising window significantly harms consistency, quality, and drift control. Qualitative comparisons in Fig. 8 also confirm that the rolling denoising window substantially reduces error accumulation, leading to fewer artifacts in long video generation.

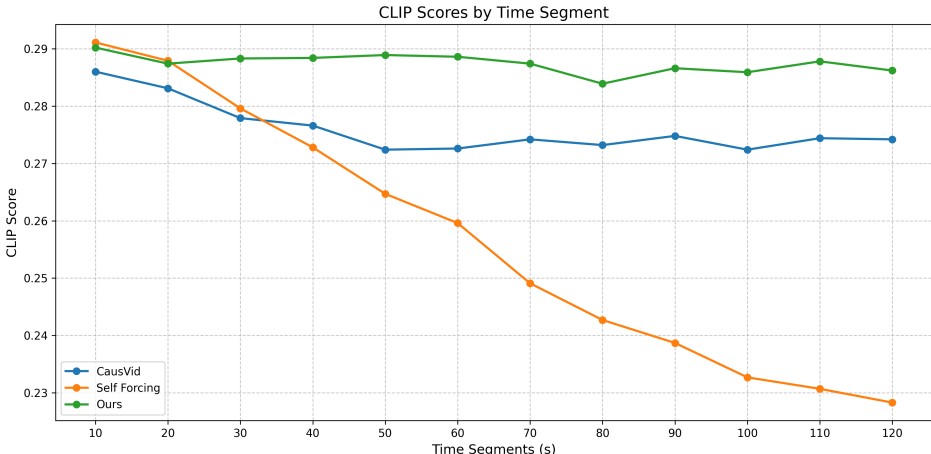

Figure 7: CLIP scores comparisons on 2-minute videos.

## D    VBENCH SCORES ACROSS ALL DIMENSIONS

We conduct a comprehensive evaluation on the full VBench benchmark (Huang et al., 2024) on 30s video clips, using all 946 prompts and covering all 16 metrics reported in Tables 4 and 5. For detailed metric definitions, we refer readers to the VBench paper. All values are computed with the official standardized evaluation scripts. Our method achieves substantial improvements in overall quality, particularly in frame-wise fidelity, and also outperforms distilled baselines on semantic scores.

Table 4: Full VBench quality evaluation on 30s videos.

| Model | Subject Consistency | Background Consistency | Temporal Flickering | Motion Smoothness | Dynamic Degree | Aesthetic Quality | Imaging Quality | Quality Score |
|---|---|---|---|---|---|---|---|---|
| CausVid (Yin et al., 2025) | 89.50 | 90.00 | **99.41** | 98.06 | 63.88 | 61.82 | 65.30 | 80.89 |
| Self Forcing (Huang et al., 2025) | 88.61 | 89.53 | 98.90 | 98.57 | **68.05** | 60.60 | 68.98 | 81.39 |
| Rolling Forcing (Ours) | **94.80** | **95.69** | 98.93 | **98.63** | 60.14 | **62.81** | **72.31** | **84.08** |

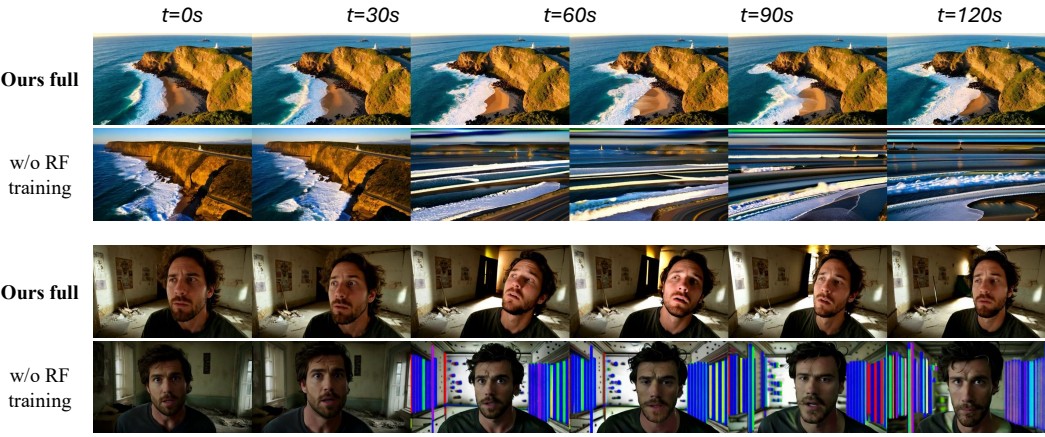

Figure 8: Ablations on Rolling Forcing. Removing the rolling denoising window significantly harms consistency, quality, and drift control.

Table 5: Full VBench semantic evaluation on 30s videos.

| Model | Object Class | Multiple Objects | Human Action | Color | Spatial Relationship | Scene | Temporal Style | Appearance Style | Overall Consistency | Semantic Score |
|---|---|---|---|---|---|---|---|---|---|---|
| CausVid (Yin et al., 2025) | 78.56 | 58.84 | 81.00 | 81.02 | 59.62 | **31.32** | 22.51 | 20.04 | 23.16 | 65.85 |
| Self Forcing (Huang et al., 2025) | 80.06 | 62.88 | **83.00** | 79.80 | 74.76 | 30.59 | **23.78** | **20.41** | **24.80** | 69.17 |
| Rolling Forcing (Ours) | **85.92** | **64.86** | 73.00 | **88.16** | **78.84** | 30.52 | 23.52 | 19.45 | 24.56 | **69.78** |

# E   ANALYSIS OF ATTENTION SINK

To better understand the mechanism of the attention sink in streaming video generation, we visualize the model's attention maps in Fig. 9. The visualizations reveal that early layers exhibit local attention patterns, while deeper layers allocate more attention weight to the global context. This indicates that the attention sink tokens function effectively by providing essential information to the denoising window.

The reasons for the attention sink phenomenon are complex. Similar to findings in LLMs (Xiao et al., 2023), we identify several possible explanations for its emergence in streaming video generation: 1) Inherent statistical difference: The latent representation of the first frame has inherently different statistics, as it is only spatially compressed, unlike subsequent frames which are also temporally compressed. 2) Information retrieval: The model learns to use the global context as a sink to retrieve important information such as color tone, white balance, etc. 3) Softmax constraint: As in LLMs, the softmax function requires attention weights to sum to one. The initial tokens may thus serve as a sink to absorb residual attention weight. This is a complex and promising research direction that extends beyond the scope of this work, and we encourage further investigation by the community.

# F   DYNAMIC ROPE

The placement of RoPE indices for the global context frames is critical. Suppose the indices of the denoising window are $i{:}i + T{-}1$, and the temporal context frames are $i - L_{\text{tem}}{:}i{-}1$. We investigate several options for assigning indices to the global context frames:

1. immediately preceding the temporal context, $i - L_{\text{tem}} - L_{\text{glo}}{:}i - L_{\text{tem}}{-}1$ (our adopted design);
2. fixed at $0{:}L_{\text{glo}}{-}1$ without dynamic RoPE adjustment;
3. overlapping with the temporal context, $i - L_{\text{glo}}{:}i{-}1$;
4. within the denoising window, within $i{:}i + T{-}1$;

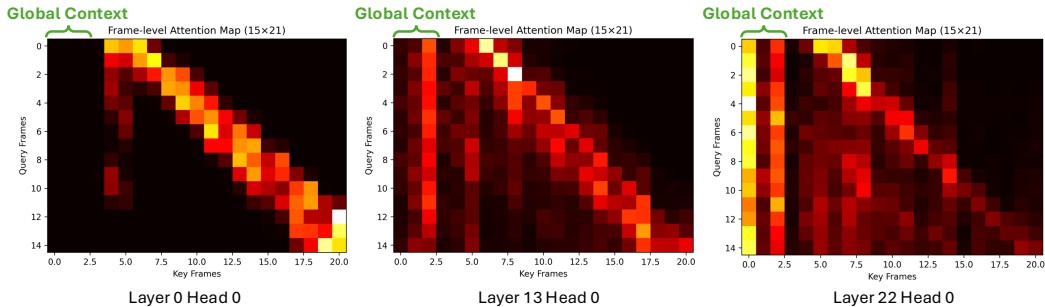

Figure 9: Visualization of the average attention logits.

5. after the denoising window, beyond $i + T$.

Empirically, option 2 produces strong "jumping" artifacts due to relative positions exceeding the trained offset range, as shown in Fig. 10. Option 3 introduces flickering, as the model confuses global and temporal contexts. Option 4 collapses into static outputs, since the generated frames are forced to replicate the global context. Option 5 induces unnatural motion, as the model attempts to converge toward the misplaced global anchor. Among these, only option 1 yields consistent videos with minimal artifacts.

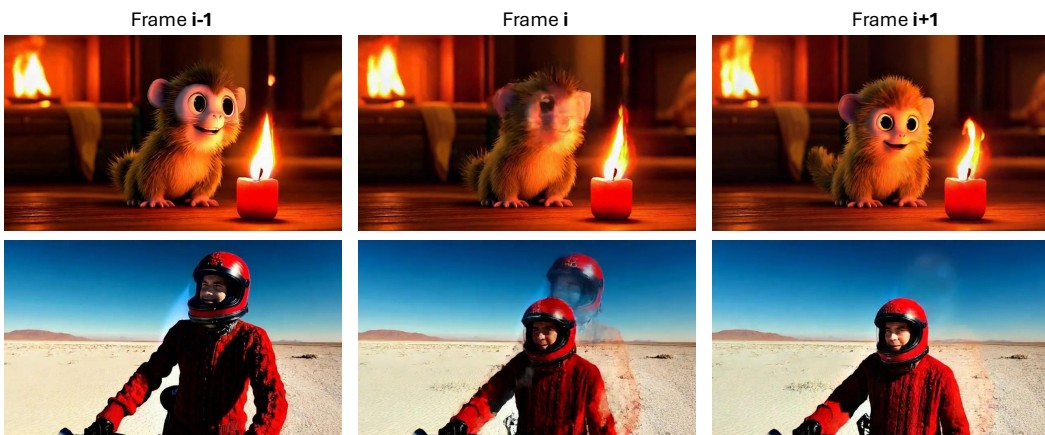

Figure 10: Fixed RoPE indices produce strong "jumping" artifacts, where the video abruptly resets to the initial frame during streaming.

## G   HUMAN PREFERENCE STUDIES

We conduct human preference studies where users select their preferred method based on error accumulation control and overall video quality. As shown in Fig. 11, our Rolling Forcing receives significantly more votes than all other methods, demonstrating its superior generation quality as perceived by human evaluators.

## H   DISCUSSIONS OF DMD TRAINING

**Training efficiency and quality of subset gradient computation.**   Although we would like to conduct an ablation study on subset gradient computation to directly assess its impact on training quality, it is infeasible to remove it entirely. The use of subset gradient computation allows Rolling Forcing to be trained on 80G GPUs; without it, the theoretical VRAM requirement would be approximately $5 \times 80 = 400G$, which is impractical to train. Nevertheless, as shown in Table 3,

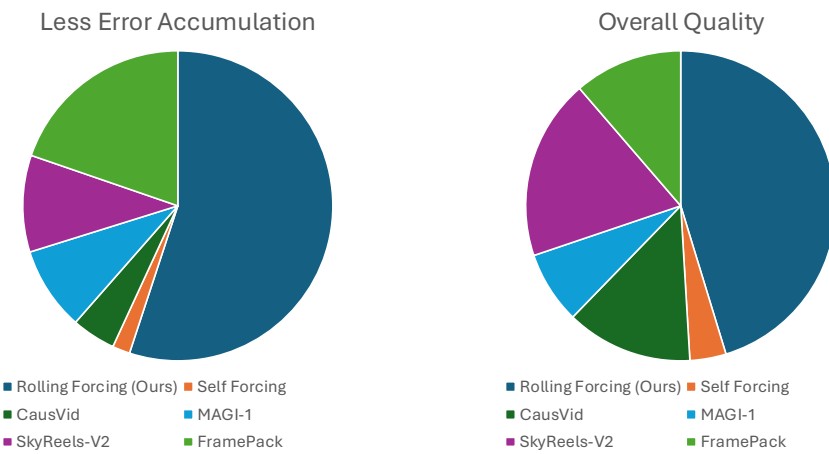

Figure 11: Human preference studies.

our method significantly outperforms the baselines, demonstrating its high quality. In terms of efficiency, our method does require more training iterations than Self Forcing (3000 vs. 600). However, the increased training time has a negligible impact on the user experience.

**Pros and cons of subset gradient computation.** The primary advantage of subset gradient computation is that it enables DMD distillation on the extended rolling denoising windows. The main disadvantage is a train-test discrepancy: during training, videos are composed of subsets of windows, whereas during inference, the final output videos are composed of the first frames from all windows.

**Mixed training strategy.** Our mixed training strategy involves alternating between Self Forcing (SF) and Rolling Forcing (RF) with equal probability. The model architecture and parameter sizes are identical for both strategies, utilizing the same global and temporal context sizes. The sole distinction lies in the generation process: SF produces videos frame-by-frame, whereas RF uses a rolling denoising window. Training is stable, as evidenced by three runs with different random seeds; both the quality drift and quality score remain approximately constant, as shown in Table 6.

Table 6: Quality Metrics Across Runs

|  | run1 | run2 | run3 |
|---|---|---|---|
| Quality Drift | 0.01 | 0.04 | 0.03 |
| Quality Score | 84.08 | 83.24 | 85.83 |

## I   LIMITATIONS

While Rolling Forcing substantially suppresses error accumulation in real-time streaming video generation, several limitations remain. First, although the global context helps stabilize long-horizon consistency, frames generated in the middle are discarded once they leave the temporal context. As a result, the model retains no memory of mid-sequence content, suggesting that incorporating more advanced memory mechanisms is a promising direction for future exploration. Second, training Rolling Forcing is computationally demanding: the enlarged attention window and the DMD loss significantly increase GPU memory usage, which may limit scalability to higher-capacity models. Developing more efficient training or distillation strategies to mitigate these costs is therefore an important avenue for future work. Third, as mentioned in Huang et al. (2025), the rolling diffusion window may increase latency in interactive applications, as future frames are partially pre-generated before the current frame is finalized. As Rolling Forcing natively supports both inference strategies, future work may consider a mixed inference strategy that interactively switches between frame-by-frame denoising during interaction and rolling denoising otherwise.

## J    BROADER SOCIETAL IMPACT

This work introduces real-time, long-horizon text-to-video generation, which can broaden access to interactive media, live storytelling, and educational tools by enabling continuous and responsive video synthesis. However, the ability to produce realistic long-duration content in real time also heightens risks of misuse, such as generating misleading live streams or amplifying harmful biases over extended outputs. We encourage future research to explore safeguards, including content filtering, bias mitigation, and responsible deployment practices, to ensure these capabilities are used for positive impact.

