# OpenReview forum: "Rolling Forcing: Autoregressive Long Video Diffusion in Real Time"
_ICLR.cc/2026/Conference — ICLR 2026 Poster_

### Official Review · Reviewer_FrmR · 2025-10-29

**Soundness:** 3
**Presentation:** 2
**Contribution:** 2
**Rating:** 4
**Confidence:** 4

**Summary:**

The paper introduces Rolling Forcing (RF), a streaming text-to-video diffusion method that reduces long-horizon error accumulation while keeping real-time throughput. Key ideas: (i) a rolling denoising window that jointly denoises several consecutive frames with bidirectional attention inside the window but emits one clean frame per forward pass; (ii) a global/temporal KV cache with dynamic RoPE so early frames act as a stable “attention-sink” anchor over minutes; and (iii) few-step DMD distillation on non-overlapping windows plus a mixed Self-Forcing (SF) objective for more natural motion. On VBench, RF improves quality and notably lowers quality drift vs. open AR baselines, with ablations for each component.

**Strengths:**

Effective drift control: Rolling window enables mutual refinement before a frame is finalized; ablations show large Quality_Drift gains.
 Long-term consistency trick: Global KV cache + dynamic RoPE keeps initial context usable without RoPE over-offset artifacts.
 Real-time demo & interactivity: Sub-second steady-state latency and prompt-switching during streaming.

**Weaknesses:**

* Attention-sink rationale needs deeper evidence: “Attention sink” comes from LMs; here it’s adapted via KV caching + dynamic RoPE. The paper shows ablations, but provides limited analysis of why the sink phenomenon should emerge in video DiTs. As written, it risks reading as a concept transplant without sufficient mechanistic support.
* Motion/scene diversity trade-off: Demos suggest mis-tuning can collapse into static outputs, RF may bias toward stillness to curb drift. More systematic tests compared to baselines on large motion and scene changes are needed, you may use like dynamic degree metric in Vbench.
* Evaluation scope: Metrics center on 30 s clips (quality drift measured 0–5 s vs. last 5 s). Given the multi-minute claim, add standardized 2–5 min quantitative runs (and a small human preference study).
* Scaling detail: Limited profiling across window/model sizes

**Questions:**

Please see weaknesses

---

> ### Author Response · Authors · 2025-11-17
>
> Dear reviewer FrmR, thank you for your review and valuable suggestions regarding our work. Below, please find our responses to your concerns.
>
> - **Weakness 1 - Analysis of Attention Sink**:
>     1. Visualizations: To better understand the mechanism of the attention sink in streaming video generation, we visualize the model's attention maps in Fig. 9 of the revised manuscript. The visualizations reveal that early layers exhibit local attention patterns, while deeper layers allocate more attention weight to the global context. This indicates that the attention sink tokens function effectively by providing essential information to the denoising window.
>
>     2. Reasons: The reasons for the attention sink phenomenon are complex. Similar to findings in LLMs, we identify several possible explanations for its emergence in streaming video generation: 1) Inherent statistical difference: The latent representation of the first frame has inherently different statistics, as it is only spatially compressed, unlike subsequent frames which are also temporally compressed. 2) Information retrieval: The model learns to use the global context as a sink to retrieve important information such as color tone, white balance, etc. 3) Softmax constraint: As in LLMs, the softmax function requires attention weights to sum to one. The initial tokens may thus serve as a sink to absorb residual attention weight. This is a complex and promising research direction that extends beyond the scope of this work, and we encourage further investigation by the community.
>
> - **Weakness 2 - Motion/Diversity trade-off**: To validate that our method's superior drift control does not come at the expense of dynamic degree, we evaluate both dynamic degree and quality drift on **2-minute​ videos**. This longer duration makes error accumulation more evident. The results in the table below confirm that our method simultaneously improves both dynamic degree and drift control compared to baselines.
>
> | Model | Dynamic Degree ↑ | Quality Drift ↓ |
> |:---|:---:|:---:|
> | CausVid | 52.08 | 3.35 |
> | Self Forcing | 51.02 | 14.4 |
> | Rolling Forcing (Ours) | **57.14** | **0.49** |
>
>
> - **Weakness 4 - Implementation Details**
>     1. Model sizes: Our method is based on the Wan2.1-T2V-1.3B model. The parameter sizes and architecture remain exactly the same as this base model.
>     2. Window sizes: We configure the KV cache and denoising window sizes as $L_\text{tem}=3$, $L_\text{glo}=3$, and $L_\text{win}=15$ latent frames.
>
>     Comprehensive implementation details are provided in Section B of the revised manuscript.

---

> ### Author Response · Authors · 2025-11-17
>
> - **Weakness 3 - Evaluation Scope**: Thank you for your suggestions. We have added the following evaluations to further validate our method. The experiments mentioned below have been included in the revised manuscript.
>     1. To better highlight the improvement in metrics, we add quantitative VBench quality comparisons with real-time baselines, i.e. CausVid and Self Forcing, over **2-minute** videos as shown in the table below and Tab. 3 in the revised manuscript. In 2-minute evaluations, Rolling Forcing demonstrates a more pronounced advantage, outperforming both baselines across nearly all quality metrics and exhibiting significantly lower quality drift.
>
>     | Model | Temp. ↑ | Subj. ↑ | Back. ↑ | Mot. ↑ | Aes. ↑ | Img. ↑ | Dyn. ↑ | Quality Drift ↓ |
>     |:---|:---:|:---:|:---:|:---:|:---:|:---:|:---:|:---:|
>     | CausVid | 96.67 | 84.69 | 89.53 | 98.04 | 62.16 | 63.62 | 52.08 | 3.35 |
>     | Self Forcing | **97.44** | 71.95 | 88.73 | 98.19 | 50.66 | 60.03 | 51.02 | 14.4 |
>     | Rolling Forcing (Ours) | 96.90 | **91.47** | **95.29** | **98.29** | **65.21** | **68.96** | **57.14** | **0.49** |
>
>     2. To further demonstrate the effectiveness of the rolling denoising window strategy, we conduct an ablation study on **2-minute** videos by removing Rolling Forcing (RF) during training, where the model is trained and inferred entirely under the frame-by-frame paradigm. As shown by the quantitative metrics in the table below and Tab. 3 in the revised manuscript, removing the rolling denoising window significantly harms consistency, quality, and drift control. Qualitative comparisons in Fig. 8 of the revised manuscript also confirm that the rolling denoising window substantially reduces error accumulation, leading to fewer artifacts in long video generation.
>
>     | Model | Temp. ↑ | Subj. ↑ | Back. ↑ | Mot. ↑ | Aes. ↑ | Img. ↑ | Dyn. ↑ | Quality Drift ↓ |
>     |:---|:---:|:---:|:---:|:---:|:---:|:---:|:---:|:---:|
>     | Ours w/o RF training | 96.87 | 84.38 | 91.93 | **98.39** | 58.61 | 66.55 | 56.75 | 1.39 |
>     | Rolling Forcing (Ours) | **96.90** | **91.47** | **95.29** | 98.29 | **65.21** | **68.96** | **57.14** | **0.49** |
>
>     3. We also analyze the CLIP score for each time segment over the **2-minute** generated videos in the table below and Fig. 7 of the revised manuscript. The results show that Rolling Forcing achieves the highest scores across segments, with the smallest score drop over time.
>
>     | Model | 0-10s | 10-20s | 20-30s | 30-40s | 40-50s | 50-60s | 60-70s | 70-80s | 80-90s | 90-100s | 100-110s | 110-120s |
>     |:---|:---:|:---:|:---:|:---:|:---:|:---:|:---:|:---:|:---:|:---:|:---:|:---:|
>     | CausVid | 0.286 | 0.2831 | 0.2779 | 0.2766 | 0.2724 | 0.2726 | 0.2742 | 0.2732 | 0.2748 | 0.2724 | 0.2744 | 0.2742 |
>     | Self Forcing | **0.2911** | **0.2879** | 0.2796 | 0.2728 | 0.2647 | 0.2596 | 0.2491 | 0.2427 | 0.2387 | 0.2327 | 0.2307 | 0.2283 |
>     | Rolling Forcing (Ours) | 0.2902 | 0.2874 | **0.2883** | **0.2884** | **0.2889** | **0.2886** | **0.2874** | **0.2839** | **0.2866** | **0.2859** | **0.2878** | **0.2862** |
>
>     4. We conduct human preference studies where users select their preferred method based on error accumulation control and overall video quality. As shown in Fig. 11 of the revised manuscript and the table below, our Rolling Forcing receives significantly more votes than all other methods, demonstrating its superior generation quality as perceived by human evaluators.
>
>     | Method                   | Less Error Accumulation | Overall Quality |
>     |--------------------------|--------------------------|-----------------|
>     | Rolling Forcing (Ours)  | **55.05%**                   | **45.28%**          |
>     | Self Forcing            | 1.83%                    | 3.77%           |
>     | CausVid                 | 4.59%                    | 13.21%          |
>     | MAGI-1                  | 8.72%                    | 7.55%           |
>     | SkyReels-V2             | 10.09%                   | 18.87%          |
>     | FramePack               | 19.72%                   | 11.32%          |

---

> > ### Comment · Reviewer_FrmR · 2025-11-28
> >
> > Thanks for the updated results and analysis. It has addressed most of my concerns. The experiment results do seem promising, though I still think the "attention sink" related claims are weird since I didn't see evidence that "attention sink" exists in the original diffusion model (not the finetuned streaming video model with global context). Nevertheless, I will consider raising my score.

---

> > > ### Author Response · Authors · 2025-11-28
> > >
> > > Thank you for your feedback and for acknowledging the updated results. We appreciate your noting the promising results of the experiments and your willingness to consider a score increase. Regarding your point about the "attention sink," we understand your perspective on the need for clearer evidence of its existence in the base model. We will revise the relevant claims in the manuscript to be more precise and focused on its observed effect within our proposed streaming framework. Thank you again for your efforts and valuable critique.

---

> ### Author Response · Authors · 2025-11-25
> **Gentle reminder regarding our rebuttal**
>
> Dear reviewer FrmR, this is a gentle reminder regarding our rebuttal. We would be very grateful if you could take a moment to review our responses when you have time. Should you have any further concerns, we are happy to provide additional clarification. Thank you again for your time and effort.

---

> ### Author Response · Authors · 2025-11-28
> **Gentle Reminder of Author-Reviewer Discussion**
>
> Dear reviewer FrmR, we hope this message finds you well. As we are nearing the end of the author-reviewer discussion period, we wanted to kindly remind you to share any final comments or feedback on the manuscript. Your input is greatly appreciated, and we would be grateful to have your perspective before concluding the discussion.
>
> Thank you again for your time and valuable contribution.

---

### Official Review · Reviewer_M4Wp · 2025-10-29

**Soundness:** 2
**Presentation:** 2
**Contribution:** 2
**Rating:** 2
**Confidence:** 4

**Summary:**

This paper introduces Rolling Forcing, a novel framework for real-time autoregressive long-video diffusion. The key idea is to mitigate error accumulation in streaming generation by performing joint denoising over a rolling window of consecutive frames rather than processing single frames sequentially. Three main contributions are proposed: (1) a rolling-window joint denoising strategy enabling mutual refinement across neighboring frames while keeping real-time throughput; (2) integration of an attention-sink mechanism that anchors initial frames as a global context to maintain long-term temporal consistency; and (3) an efficient few-step distillation algorithm over non-overlapping windows that conditions on self-generated histories to reduce exposure bias.
Experimental results show that Rolling Forcing can stream multi-minute videos at 16 FPS on a single GPU with minimal drift, achieving state-of-the-art performance on the VBench benchmark and outperforming prior autoregressive baselines such as Self Forcing and CausVid in both visual quality and temporal stability.

**Strengths:**

1. Rolling forcing is a robust and effective framework. Components like progressively increasing noise levels, attention sink, and few-step distillation enhance long video generation.
2. Compared to baselines like causvid and self-forcing, rolling forcing significantly reduces error accumulation and extends video length while maintaining content consistency.

**Weaknesses:**

The weaknesses primarily stem from the novelty and experiments, as outlined below:
1. The proposed components are not entirely novel, as they have been used in video generation. For example, progressively increasing noise is used in Rolling Diffusion[1], MAGI-1[2], and FIFO-Diffusion[3]. Attention sink has been applied in LLMs[4], and step distillation is utilized in self-forcing[5]. This paper seems to integrate these components into an autoregressive video generation paradigm.
2. The paper does not compare its approach to related works in autoregressive video generation, such as FramePack[6], which it cites.
3. The training length in the paper is set to 27, with a KV cache length of 24 frames. It raises questions about how the attention sink performs with such a short training length.
4. In Table 2, the second row (without RF training) achieves strong performance, though still lower than rolling forcing. This raises the question of the key benefits of RF training.

[1] Rolling Diffusion Models.
[2] MAGI-1: Autoregressive Video Generation at Scale.
[3] FIFO-Diffusion: Generating Infinite Videos from Text without Training.
[4] When Attention Sink Emerges in Language Models
[5] Self Forcing: Bridging the Train-Test Gap in Autoregressive Video Diffusion
[6] Frame Context Packing and Drift Prevention in Next-Frame-Prediction Video Diffusion Models.

**Questions:**

1. What motivates the focus on long video generation, and how is this challenge tackled?
2. How does this paper's contribution distinguish itself from the limitations of related works?
3. How do the results compare to FramePack?

---

> ### Author Response · Authors · 2025-11-17
>
> Dear reviewer M4Wp, thank you for your review and valuable suggestions regarding our work. Below, please find our responses to your concerns.
>
> - **Weakness 1 - Novelty**: We thank the reviewer for their perspective. Our work is inspired by several great previous works, all of which we have cited and discussed in the original manuscript. We would like to highlight the following key novel contributions:
>     1. **First Application of Attention Sink to Video Generation**: To the best of our knowledge, we are the first to adapt attention sink for video generation, which is crucial for managing long video generation.
>     2. **Novel DMD Distillation Framework**: We are the first to apply DMD on rolling windows. Furthermore, our design of the DMD loss for non-overlapping windows is an entirely new contribution.
>     3. **State-of-the-Art Performance**: Our method achieves real-time throughput with minimal error accumulation over multi-minute videos, representing a significant advancement for long video generation.
>     4. **Fundamentally Different Technical Approach**: While other methods also employ varying noise schedules, our technique is fundamentally distinct. For instance: 1) Rolling Diffusion Models is a pre-training method using a Diffusion Forcing objective. 2) FIFO is a training-free method that relies on a complex denoising strategy. 3) MAGI is also a pre-training method that uses causal attention within its windows. In contrast, our approach is based on a distillation framework and utilizes bidirectional attention within the rolling windows, enabling better drift control and real-time throughput.
>
>
> - **Weakness 2 and Question 3 - Compare with FramePack**: We have added comparisons with the **non-streaming image-to-video** method FramePack, which generates long videos in a non-sequential order, in the table below and in Tab. 1 of the revised manuscript. We use the first frame generated by our method as the image input for FramePack. Notably, our Rolling Forcing method outperforms FramePack in quality, consistency, and drift control, despite FramePack being a 10 times larger image-to-video model. As illustrated in Fig. 4 of the revised manuscript, FramePack still exhibits severe error accumulation in long video generation, often resulting in static outputs. It is important to note that since FramePack generates videos in reverse order, this error accumulation is most pronounced at the beginning of the videos. We also provide video comparisons, which can be downloaded via this anonymous link: https://anonymous.4open.science/r/Anonymous_Rolling_Forcing_Webpage_Submission_Number_4445/
>
> | Model | #Params | Throughput (FPS) ↑ | Latency (s) ↓ | Temporal Flickering ↑ | Subject Consistency ↑ | Background Consistency ↑ | Motion Smoothness ↑ | Aesthetic Quality ↑ | Imaging Quality ↑ | Quality Drift ↓ |
> |:---|:---:|:---:|:---:|:---:|:---:|:---:|:---:|:---:|:---:|:---:|
> | FramePack | 13B | 0.92 | 65 | **99.26** | 91.65 | 93.55 | **99.03** | 59.93 | 65.20 | 3.45 |
> | Rolling Forcing (Ours) | 1.3B | **15.79** | **0.76** | 97.61 | **92.80** | **93.71** | 98.70 | **62.39** | **70.75** | **0.01** |
>
>
> - **Weakness 3 - Length of KV Cache and Effectiveness of Attention Sink**:
>     1. Length of KV Cache: There is a fact error we'd like to point out, as mentioned in Sec. B, the KV cache length is $L_\text{tem}+L_\text{glo}=6$ instead of 24. The maximum distance between the attention sink tokens and denoising window during training is 27, which is sufficient for attention sink training.
>     2. Effectiveness of Attention Sink: We provide two experiments to validate the effectiveness of attention sink: 1) We provide an ablation study of attention sink in Tab. 2 and Fig. 5 in the original manuscript and the table below, in which attention sink clearly improves drift control. 2) In Sec. F of the revised manuscript, we visualize the model's attention maps. The visualizations reveal that early layers exhibit local attention patterns, while deeper layers allocate more attention weight to the attention sink tokens. This indicates that attention sink functions effectively by providing essential information to the denoising window.
> | Model | Temp. ↑ | Subj. ↑ | Back. ↑ | Mot. ↑ | Aes. ↑ | Img. ↑ | Quality Drift ↓ |
> |:---|:---:|:---:|:---:|:---:|:---:|:---:|:---:|
> | w/o attention sink | 97.53 | 83.22 | 87.99 | 98.56 | 58.99 | 67.30 | 4.63 |
> | Ours full | **97.61** | **92.80** | **93.71** | **98.70** | **62.39** | **70.75** | **0.01** |

---

> ### Author Response · Authors · 2025-11-17
>
> - **Weakness 4 - Effectiveness of RF Training**: To further demonstrate the effectiveness of the rolling denoising window strategy, we conduct an ablation study on **2-minute** videos by removing Rolling Forcing (RF) during training, where the model is trained and inferred entirely under the frame-by-frame paradigm. As shown by the quantitative metrics in the table below and Tab. 3 in the revised manuscript, removing the rolling denoising window significantly harms consistency, quality, and drift control. Qualitative comparisons in Fig. 8 of the revised manuscript also confirm that the rolling denoising window substantially reduces error accumulation, leading to fewer artifacts in long video generation.
>
> | Model | Temp. ↑ | Subj. ↑ | Back. ↑ | Mot. ↑ | Aes. ↑ | Img. ↑ | Dyn. ↑ | Quality Drift ↓ |
> |:---|:---:|:---:|:---:|:---:|:---:|:---:|:---:|:---:|
> | Ours w/o RF training | 96.87 | 84.38 | 91.93 | **98.39** | 58.61 | 66.55 | 56.75 | 1.39 |
> | Rolling Forcing (Ours) | **96.90** | **91.47** | **95.29** | 98.29 | **65.21** | **68.96** | **57.14** | **0.49** |
>
>
> - **Question 1 - What motivates the focus on long video generation, and how is this challenge tackled?** We focus on streaming long video generation as it is the fundamental component in interactive world models and neural game engine. However, most existing work suffers from severe error accumulation that often significantly degrades the generated stream videos over long horizons. We tackle this challenge by: First, we introduce a rolling-window joint denoising technique that processes multiple frames in a single forward pass, enabling mutual refinement while preserving real-time latency. Second, we introduce the attention sink mechanism into the streaming video generation task, a pioneering effort that enables caching the initial frames as consistent global context for long-term coherence in video generation. Third, we design an efficient training algorithm that operates on non-overlapping windows and conditions on self-generated histories, enabling few-step distillation over extended denoising windows and concurrently mitigating exposure bias.
>
> - **Question 2 - How does this paper's contribution distinguish itself from the limitations of related works?** Rolling Forcing performs real-time streaming text-to-video generation at 16 fps on a single GPU and is capable of producing multi-minute-long videos with minimal error accumulation. Other methods like Self Forcing and CausVid suffer from severe error accumulation while FramePack, Skyreels are not real-time.

---

> ### Comment · Reviewer_M4Wp · 2025-11-24
>
> Thank you for the detailed response. After carefully reviewing each point, I still have the following concern:
>
> While the paper suggests the non-overlapping window loss is the main contribution for reducing error accumulation in streaming video generation (compared to self-forcing), the response to Weakness 3 indicates that the attention sink is actually the primary factor addressing quality drift (w. attention sink 0.01 vs. wo attention sink 4.61). I remain confused about the distinct roles of these two components.

---

> > ### Author Response · Authors · 2025-11-25
> >
> > Dear reviewer M4Wp, thank you for your follow-up message. Regarding your concern, we would like to clarify that the attention sink and the rolling denoising window are the two main components that contribute to reducing error accumulation. The purpose of the non-overlapping window training strategy is to reduce memory consumption, allowing us to perform DMD distillation on the extended rolling denoising window.
> >
> > Regarding the distinct roles of the attention sink and the rolling denoising window, both contribute to reducing error accumulation but through different mechanisms. Below is a table that elaborates on their differences.
> >
> >
> > | Component                | Mechanism                          | Effect                                                                 |
> > | :----------------------- | :--------------------------------- | :--------------------------------------------------------------------- |
> > | **Attention sink**       | Maintains a persistent global anchor | Improves long-term consistency, thereby reducing drift.               |
> > | **Rolling denoising window** | Enables bidirectional attention across frames | Allows for mutual refinement between frames, which helps correct errors. |
> >
> > Both components are essential for generating high-quality multi-minute videos. **Please refer to Fig. 8 in the revised manuscript**, which clearly shows that removing the rolling denoising window would cause severe error accumulation during the generation of multi-minute videos.
> >
> > Should you have any further concerns, we are happy to provide additional clarification.

---

> > ### Author Response · Authors · 2025-11-28
> > **Gentle Reminder of Author-Reviewer Discussion**
> >
> > Dear reviewer M4Wp, thank you for your review and for replying to our rebuttal. As we near the end of the author-reviewer discussion, we would like to kindly remind you to share any final comments or feedback on the manuscript. Your input is greatly appreciated. Should you have any further concerns, we are happy to provide additional clarification.

---

> ### Comment · Reviewer_M4Wp · 2025-11-28
>
> Thank you for the detailed response, which addresses most of my concerns. I still recommend investigating why attention sink significantly reduces error accumulation. Overall, I think rolling forcing is promising framework for streaming long video generation, and I may consider raising my score after the rebuttal.

---

> ### Author Response · Authors · 2025-11-28
>
> Thank you for your thoughtful feedback and for the positive comments on the potential of our work. We agree that investigating the reason behind attention sink's effectiveness is a valuable direction and have added an analysis of this point to the revised manuscript in Sec. F. Thank you again for your time and effort.

---

### Official Review · Reviewer_zgDS · 2025-10-30

**Soundness:** 4
**Presentation:** 4
**Contribution:** 4
**Rating:** 8
**Confidence:** 5

**Summary:**

This paper proposed a method to generate long video without drifting error and enables streaming video. Instead of causal diffusion, it uses progressive noise level for denoising. It utilizes attention sink to keep long term video consistency and quality. Finally it proposed a few-step distillation method over extended window to mitigate exposure bias conditioned on self-generated histories.

**Strengths:**

- introducing rolling diffusion, i.e., progressive noise level for multiple video latents, into self-forcing pipeline to achieve smoother and lower error drifting denoising framework.
- history frame use causal attention so as to be saved in KV cache to save inference compute.
- adopted attention sink with a few global context at the beginning of the sequence. Adjust RoPE dynamically to avoid PE index out of range.
- novel design of DMD loss on progressive noised latent to save computation cost.
- the error drifting problem has been improved significantly.

**Weaknesses:**

- the design of using a subset of backpropogated gradients for DMD loss training alleviate the memory and compute cost. But has the training quality and efficiency been affected at the same time? For example, it is mentioned that the original SF training is still needed to regularize the training. It would be helpful to have a discussion about advantage and disadvantage of it since this is one of the core contribution.
- the overall design resolves the error drifting significantly, but other metrics, e.g., fps, latency, VBench score, are not improved or improved marginally.

**Questions:**

- How does attention sink perform for long video with context dramatically changed?
- It is not clear how the mixed training strategy (SF + Rolling Forcing) performed practically. Besides, is this training strategy stable? It would be helpful to add a more detailed discussion of it.
- Since RoPE is very sensitive, does dynamically changing RoPE for attention sink bring any artifacts or instability of the output videos?

---

> ### Author Response · Authors · 2025-11-17
>
> Dear reviewer zgDS, thank you for your review and valuable suggestions regarding our work. Below, please find our responses to your concerns.
>
> -  **Weakness 1 - Subset Gradients**: Thank you for your suggestions regarding additional discussions. We have added the following discussions to Sec. I of the revised manuscript.
>     1. Training efficiency and quality of subset gradient computation: Although we would like to conduct an ablation study on subset gradient computation to directly assess its impact on training quality, it is infeasible to remove it entirely. The use of subset gradient computation allows Rolling Forcing to be trained on 80G GPUs; without it, the theoretical VRAM requirement would be approximately 5 × 80 = 400G, which is impractical to train. Nevertheless, as shown in Table 3 of the revised manuscript, our method significantly outperforms the baselines, demonstrating its high quality. In terms of efficiency, our method does require more training iterations than Self Forcing (3000 vs. 600). However, the increased training time has a negligible impact on the user experience.
>     2. Pros and cons of subset gradient computation: The primary advantage of subset gradient computation is that it enables DMD distillation on the extended rolling denoising windows. The main disadvantage is a train-test discrepancy: during training, videos are composed of subsets of windows, whereas during inference, the final output videos are composed of the first frames from all windows.
>
> - **Weakness 2 - Limited Metric Improvement**: To better highlight the improvement in metrics, we add quantitative VBench quality comparisons with real-time baselines, i.e. CausVid and Self Forcing, over **2-minute** videos as shown in the table below and Tab. 3 in the revised manuscript. In 2-minute evaluations, Rolling Forcing demonstrates a more pronounced advantage, outperforming both baselines across nearly all quality metrics and exhibiting significantly lower quality drift.
>
> | Model | Temp. ↑ | Subj. ↑ | Back. ↑ | Mot. ↑ | Aes. ↑ | Img. ↑ | Dyn. ↑ | Quality Drift ↓ |
> |:---|:---:|:---:|:---:|:---:|:---:|:---:|:---:|:---:|
> | CausVid | 96.67 | 84.69 | 89.53 | 98.04 | 62.16 | 63.62 | 52.08 | 3.35 |
> | Self Forcing | **97.44** | 71.95 | 88.73 | 98.19 | 50.66 | 60.03 | 51.02 | 14.4 |
> | Rolling Forcing (Ours) | 96.90 | **91.47** | **95.29** | **98.29** | **65.21** | **68.96** | **57.14** | **0.49** |
>
>
>
> - **Question 1 - Dramatic Changes**: We have tested our method on scenarios with significant changes, and additional examples are provided in Fig. 6 of the revised manuscript. Our method is capable of handling various types of substantial changes, including subject switch, subject introduction, scene switch, and action switch. In our tests, the attention sink helps stabilize the generation process but does not necessarily restrict the content or changes in the video stream. Elements such as the environment, action, and subject can all be modified or introduced. Additional video examples can be downloaded via this anonymous link: https://anonymous.4open.science/r/Anonymous_Rolling_Forcing_Webpage_Submission_Number_4445/
>
> - **Question 2 - Mixed Training**: Our mixed training strategy involves alternating between Self Forcing (SF) and Rolling Forcing (RF) with equal probability. The model architecture and parameter sizes are identical for both strategies, utilizing the same global and temporal context sizes. The sole distinction lies in the generation process: SF produces videos frame-by-frame, whereas RF uses a rolling denoising window. Training is stable, as evidenced by three runs with different random seeds; both the quality drift and quality score remain
> approximately constant, as shown in the table below or Tab. 6 of the revised manuscript.
>
> |  | **run1** | **run2** | **run3** |
> |:---|:---:|:---:|:---:|
> | Quality Drift | 0.01 | 0.04 | 0.03 |
> | Quality Score | 84.08 | 83.24 | 85.83 |
>
> - **Question 3 - Dynamic RoPE**: Dynamically changing the RoPE indices of the sink tokens does not introduce artifacts or instability. On the contrary, using a fixed RoPE produces strong "jumping" artifacts, where the video abruptly resets to the initial frame during streaming. We provide additional visualizations of these fixed RoPE artifacts in Fig. 10 of the revised manuscript.

---

> > ### Comment · Reviewer_zgDS · 2025-11-28
> >
> > Thanks authors for the response. It has addressed all my concerns. The new evaluation on 2-min video has significant improvement which should be highlighted. Please include all the promised changes in the final manuscript.

---

> > > ### Author Response · Authors · 2025-11-28
> > >
> > > Thank you for your positive feedback. We are glad to hear that our responses have addressed your concerns. We have added the additional results, including the new evaluation on the 2-minute videos, to the revised manuscript as promised.

---

### Official Review · Reviewer_PmgV · 2025-11-01

**Soundness:** 4
**Presentation:** 3
**Contribution:** 3
**Rating:** 6
**Confidence:** 4

**Summary:**

This paper proposes "Rolling Forcing," a method designed to alleviate error accumulation in long-horizon video generation. It uses a rolling denoising window and introduces an attention sink mechanism to improve temporal coherence and achieve real-time rendering through few-step distillation. The paper includes detailed pseudocode and extensive engineering contributions, clearly explaining the training process.

**Strengths:**

- The proposed attention sink clearly helps alleviate error accumulation issues, as demonstrated in Figure 4.
- Few-step distillation allows real-time rendering, making the approach practical for streaming video generation.
- The paper includes substantial engineering work, and the provided pseudocode clearly describes the training process in detail.

**Weaknesses:**

- A significant limitation of the proposed approach is that the use of the attention sink and Rolling Forcing may not generalize well to long videos with drastically changing content. This setup could restrict substantial changes in video content, causing generated videos to appear as repetitive sequences or stitched clips (as evident in all the provided supplementary videos). Additionally, continuity issues arise when key frames do not provide rich enough information.
- The evaluation is somewhat limited, relying only on metrics from VBench. A broader evaluation would strengthen the claims.

**Questions:**

- Could the authors discuss how well their approach handles scenarios with significant changes in video content?

---

> ### Author Response · Authors · 2025-11-17
>
> Dear reviewer PmgV, thank you for your review and valuable suggestions regarding our work. Below, please find our responses to your concerns.
>
> - **Weakness 1 and Question 1 - Limited Changes**: We have tested our method on scenarios with significant changes, and additional examples are provided in Fig. 6 of the revised manuscript. Our method is capable of handling various types of substantial changes, including subject switch, subject introduction, scene switch, and action switch. In our tests, the attention sink helps stabilize the generation process but does not necessarily restrict the content or changes in the video stream. Elements such as the environment, action, and subject can all be modified or introduced. Additional video examples can be downloaded via this anonymous link: https://anonymous.4open.science/r/Anonymous_Rolling_Forcing_Webpage_Submission_Number_4445/
>
>
> - **Weakness 2 - Limited Evaluations**: Thank you for your suggestions. We have added the following evaluations to further validate our method. The experiments mentioned below have been included in the revised manuscript.
>
>     1. To better highlight the improvement in metrics, we add quantitative VBench quality comparisons with real-time baselines, i.e. CausVid and Self Forcing, over **2-minute** videos as shown in the table below and Tab. 3 in the revised manuscript. In 2-minute evaluations, Rolling Forcing demonstrates a more pronounced advantage, outperforming both baselines across nearly all quality metrics and exhibiting significantly lower quality drift.
>
>     | Model | Temp. ↑ | Subj. ↑ | Back. ↑ | Mot. ↑ | Aes. ↑ | Img. ↑ | Dyn. ↑ | Quality Drift ↓ |
>     |:---|:---:|:---:|:---:|:---:|:---:|:---:|:---:|:---:|
>     | CausVid | 96.67 | 84.69 | 89.53 | 98.04 | 62.16 | 63.62 | 52.08 | 3.35 |
>     | Self Forcing | **97.44** | 71.95 | 88.73 | 98.19 | 50.66 | 60.03 | 51.02 | 14.4 |
>     | Rolling Forcing (Ours) | 96.90 | **91.47** | **95.29** | **98.29** | **65.21** | **68.96** | **57.14** | **0.49** |
>
>     2. To further demonstrate the effectiveness of the rolling denoising window strategy, we conduct an ablation study on **2-minute** videos by removing Rolling Forcing (RF) during training, where the model is trained and inferred entirely under the frame-by-frame paradigm. As shown by the quantitative metrics in the table below and Tab. 3 in the revised manuscript, removing the rolling denoising window significantly harms consistency, quality, and drift control. Qualitative comparisons in Fig. 8 of the revised manuscript also confirm that the rolling denoising window substantially reduces error accumulation, leading to fewer artifacts in long video generation.
>
>     | Model | Temp. ↑ | Subj. ↑ | Back. ↑ | Mot. ↑ | Aes. ↑ | Img. ↑ | Dyn. ↑ | Quality Drift ↓ |
>     |:---|:---:|:---:|:---:|:---:|:---:|:---:|:---:|:---:|
>     | Ours w/o RF training | 96.87 | 84.38 | 91.93 | **98.39** | 58.61 | 66.55 | 56.75 | 1.39 |
>     | Rolling Forcing (Ours) | **96.90** | **91.47** | **95.29** | 98.29 | **65.21** | **68.96** | **57.14** | **0.49** |
>
>     3. We also analyze the CLIP score for each time segment over the **2-minute** generated videos in the table below and Fig. 7 of the revised manuscript. The results show that Rolling Forcing achieves the highest scores across segments, with the smallest score drop over time.
>
>     | Model | 0-10s | 10-20s | 20-30s | 30-40s | 40-50s | 50-60s | 60-70s | 70-80s | 80-90s | 90-100s | 100-110s | 110-120s |
>     |:---|:---:|:---:|:---:|:---:|:---:|:---:|:---:|:---:|:---:|:---:|:---:|:---:|
>     | CausVid | 0.286 | 0.2831 | 0.2779 | 0.2766 | 0.2724 | 0.2726 | 0.2742 | 0.2732 | 0.2748 | 0.2724 | 0.2744 | 0.2742 |
>     | Self Forcing | **0.2911** | **0.2879** | 0.2796 | 0.2728 | 0.2647 | 0.2596 | 0.2491 | 0.2427 | 0.2387 | 0.2327 | 0.2307 | 0.2283 |
>     | Rolling Forcing (Ours) | 0.2902 | 0.2874 | **0.2883** | **0.2884** | **0.2889** | **0.2886** | **0.2874** | **0.2839** | **0.2866** | **0.2859** | **0.2878** | **0.2862** |
>
>     4. We conduct human preference studies where users select their preferred method based on error accumulation control and overall video quality. As shown in Fig. 11 of the revised manuscript and the table below, our Rolling Forcing receives significantly more votes than all other methods, demonstrating its superior generation quality as perceived by human evaluators.
>
>     | Method                   | Less Error Accumulation | Overall Quality |
>     |--------------------------|--------------------------|-----------------|
>     | Rolling Forcing (Ours)  | **55.05%**                   | **45.28%**          |
>     | Self Forcing            | 1.83%                    | 3.77%           |
>     | CausVid                 | 4.59%                    | 13.21%          |
>     | MAGI-1                  | 8.72%                    | 7.55%           |
>     | SkyReels-V2             | 10.09%                   | 18.87%          |
>     | FramePack               | 19.72%                   | 11.32%          |

---

> ### Author Response · Authors · 2025-11-25
> **Gentle reminder regarding our rebuttal**
>
> Dear reviewer PmgV, this is a gentle reminder regarding our rebuttal. We would be very grateful if you could take a moment to review our responses when you have time. Thank you again for your time and effort.

---

> > ### Comment · Reviewer_PmgV · 2025-11-25
> >
> > Thank the authors for the detailed rebuttal and additional experiments provided.
> >
> > While the examples in Figure 6 partially address my concerns about handling significant content changes, there remains room for improvement in demonstrating the method's generalization to more drastic content variations. Additionally, although the newly added CLIP scores are informative, they primarily reflect text-video alignment rather than broader perceptual quality or temporal coherence.
> >
> > However, considering the technical strength and practical relevance of the work, and noting that some other reviews might not fully capture these positive aspects, I have decided to increase my rating to 8 to better balance the overall evaluation.

---

> > > ### Author Response · Authors · 2025-11-26
> > >
> > > Thank you for your thoughtful feedback and for the positive decision to increase your rating. We appreciate your acknowledgment of the technical strength and practical relevance of our work.
> > >
> > > We also thank you for the constructive suggestions regarding the demonstration of generalization to more drastic content variations and the evaluation of perceptual quality. We will add a note in the revised manuscript to discuss these points as valuable directions for future research.
> > >
> > > Your insightful comments have certainly helped to improve our paper.

---

### Author Response · Authors · 2025-12-01
**Summary of Rebuttal (part1/2)**

To the Area Chair,

We are sorry to learn about the recent information leak on OpenReview and the resulting impact on the reviewing process of ICLR. We understand that this unexpected incident has introduced additional workload, and we would like to express our sincere appreciation to the AC for the extra effort required during this challenging situation.
To help reduce the burden on the AC, we summarize below the key progress that was explicitly acknowledged by the reviewers during the two-week rebuttal period prior to the information leak incident.

We are greatly encouraged that **all four reviewers acknowledged that our detailed responses and additional experiments have addressed their primary concerns. Each reviewer has indicated a positive shift in their assessment, with three explicitly stating their intention to raise their scores**. Below is a concise summary of each reviewer's journey.

## Reviewer PmgV (Initial Rating: 6 → Final Stance: Raise to 8)
- *Initial Concerns*:
    1. The proposed method may not generalize well to long videos with drastically changing content, potentially leading to repetitive or stitched appearances.
    2. The evaluation was somewhat limited, relying primarily on VBench metrics.
- *Our Response*:
    1. We provided additional examples (Fig. 6) demonstrating our method's capability to handle significant content changes, including subject and scene switches.
    2. We expanded the evaluation with quantitative comparisons on 2-minute videos, an ablation study, temporal CLIP score analysis, and human preference studies to comprehensively validate our method's performance and reduced error accumulation.
- *Reviewer's Final Feedback*: Acknowledged that the examples "partially addressed" the concern and found the new CLIP scores "informative." Stated that considering the "technical strength and practical relevance," they decided to **increase their rating to 8** to better reflect the work's quality and balance the overall evaluation.

## Reviewer zgDS (Initial Rating: 8 → Final Stance: Maintain 8, Concerns Addressed)
- *Initial Concerns*:
    1. The subset gradient computation for DMD loss may affect training quality/efficiency, requiring further discussion of its trade-offs.
    2. While error drifting improved significantly, other metrics showed only marginal gains.
    3. Questions about attention sink performance with dramatic content changes, stability of mixed training, and potential artifacts from dynamic RoPE adjustments.
- *Our Response*:
    1. Added discussion on subset gradient trade-offs: it enables feasible training (400G→80G VRAM) despite train-test discrepancy, with results demonstrating superior quality.
    2. Provided new 2-minute video evaluations showing pronounced VBench metric improvements and significantly lower quality drift versus baselines.
    3. Clarified attention sink handles content changes effectively (Fig. 6), demonstrated mixed training stability via multi-seed experiments, and showed dynamic RoPE prevents artifacts (fixed RoPE causes jumping).
- *Reviewer's Final Feedback*: Concluded, "Thanks authors for the response. **It has addressed all my concerns.** The new evaluation on 2-min video has significant improvement which should be highlighted." They were satisfied with our revisions.

---

> ### Author Response · Authors · 2025-12-01
> **Summary of Rebuttal (part2/2)**
>
> ## Reviewer M4Wp (Initial Rating: 2 → Final Stance: Willing to Raise Score)
> - *Initial Concerns*:
>     1. The novelty of the proposed components was questioned.
>     2. The paper lacked comparison with the cited work FramePack.
>     3. There was confusion regarding the KV cache length (incorrectly stated as 24) and the effectiveness of the attention sink with a short training length.
>     4. The distinct roles and relative importance of the rolling denoising window and the attention sink mechanism in reducing error accumulation were unclear.
> - *Our Response*:
>     1. We clarified the novel aspects of our work: the first application of attention sink to video generation, a novel DMD distillation framework for non-overlapping windows, and a fundamentally different technical approach that achieves state-of-the-art performance.
>     2. We added comprehensive comparisons with FramePack in the revised manuscript (Tab. 1, Fig. 4), showing our method's superiority in quality, consistency, and drift control despite using a smaller model.
>     3. We corrected the factual error regarding the KV cache length (which is 6) and provided ablation studies (Tab. 2) and attention map visualizations (Sec. F) to validate the effectiveness of the attention sink.
>     4. We explained that both the rolling denoising window (enabling bidirectional refinement) and the attention sink (providing a long-term anchor) are essential and complementary components for reducing error accumulation, as demonstrated in the new ablation study on 2-minute videos (Tab. 3, Fig. 8).
> - *Reviewer's Final Feedback*: After further discussion, the reviewer acknowledged that our responses **addressed most concerns**, stating, "I think rolling forcing is promising framework... and I may consider **raising my score** after the rebuttal." In their final comment, they confirmed this position.
>
>
> ## Reviewer FrmR (Initial Rating: 4 → Final Stance: Willing to Raise Score)
> - *Initial Concerns*:
>     1. The rationale for the attention sink mechanism required deeper evidence.
>     2. There was a potential trade-off where superior drift control might come at the expense of motion and scene diversity, necessitating more systematic evaluation using metrics like dynamic degree.
>     3. The evaluation scope was limited to 30-second clips; expanded quantitative runs on 2-5 minute videos and a human preference study were recommended.
>     4. More detailed profiling of window and model sizes was requested.
> - *Our Response*:
>     1. We provided attention map visualizations (Fig. 9) and a discussion of potential reasons for the attention sink's emergence in video generation, such as inherent statistical differences and softmax constraints.
>     2. We added evaluations on 2-minute videos, demonstrating that our method simultaneously improves both dynamic degree and quality drift compared to baselines.
>     3. We expanded the evaluation with comprehensive 2-minute VBench metrics, temporal CLIP score analysis, and a human preference study, all included in the revised manuscript.
>     4. We clarified the implementation details, including model and window sizes, in Section B of the manuscript.
> - *Reviewer's Final Feedback*: "Thanks for the updated results and analysis. **It has addressed most of my concerns**. The experiment results do seem promising... I will consider **raising my score**."
>
> ## Conclusion
> The reviewer discussions demonstrate a clear consensus: our rebuttal successfully alleviated the initial concerns. The key strengths of our work: real-time, long-video generation with minimal error accumulation, backed by extensive quantitative, qualitative, and human evaluations, were validated through this process. We are grateful for the reviewers' insightful comments, which have undoubtedly strengthened the paper.
>
> We hope the Area Chair will consider the **overwhelmingly positive trajectory** of the reviews and the reviewers' collective endorsement of the revised manuscript. Thank you for your time and effort.
>
> Sincerely,
>
> The Authors

---

### Meta-Review · Area_Chair_JUdi · 2026-01-07

**Summary:**

This paper presents Rolling Forcing for real-time streaming long video generation. The experimental results demonstrate that Rolling Forcing achieves real-time generation while producing multi-minute videos with substantially reduced quality drift. All reviewers acknowledged during the discussion that their primary concerns were partically addressed by the rebuttal. The AC recognizes that the experimental results are strong and comprehensive and the authors well addressed reviewers’ concern with more evaluations and human preference studies.

**Reviewer Concerns:**

Addressed concerns: (1) generalization to drastic content changes. The authors demonstrated that the method can handle subject switches, scene transitions, and action changes. (2) the concern about limited evaluation scope. The authors provided CLIP score analysis and human preference studies.

Outstanding concerns: the mechanistic of why attention sink emerges in video diffusion transformers remains somewhat speculative.

**Reviewer Scores:**

Reviewer M4Wp questions the novelty of components, missing FramePack comparison, and KV cache length. The authors' responses well addressed these concerns. The reviewer might raise the score.

Reviewer FrmR initially rated the paper at 4, requesting deeper evidence for attention sink, more evaluation of motion/diversity trade-offs. The authors' responses partially addressed these concerns. The reviewer might raise the score, but there are remaining questions about the attention sink mechanism.

---

### Decision · Program_Chairs · 2026-01-26

Accept (Poster)